# Stability of the marine nitrogen cycle over the past 165 million years

Linda V. Godfrey[1] ✉, Anne Willem Omta[2], Eli Tziperman [3], Xiang Li [4], Yongyun Hu [4,5] & Paul G. Falkowski[1,6]

Nitrogen and phosphorus are the two macro-nutrients that limit biological productivity in the ocean. While the supply of P depends on geological processes, N is biologically supplied from an inexhaustible atmospheric source, but can be limited by micro-nutrients, especially iron. Here we present a record of N and C isotopes over the past 165 Ma in marine sediments to address feedbacks between the N-cycle and productivity. Over most of the last 165 Myr, the fixed N averaged +3.2‰, (−2 and +9‰), but higher in distal areas of the ocean due to limited vertical mixing. Using an isotope box model and a coupled climate model we show that this is caused by winds that induce upwelling changing due to continental meander. Upwelling along low latitude east-west orientated Tethyan coastlines results in low $\delta^{15}N$, while upwelling along narrow N-S coastlines as it does today, results in high $\delta^{15}N$ due to denitrification.

In aquatic ecosystems, the availability of two macronutrients, nitrogen (dissolved inorganic N, nitrate and ammonium) and phosphorus (dissolved inorganic P), exerts control on the productivity and biomass of primary producers[1,2]. These two elements have contrasting biogeochemical cycles. Phosphate is released and consumed on land by rock weathering and soil formation, and transferred to the oceans by rivers, and to a lesser extent, windblown dust[3]. P is lost from the ocean through burial associated with biological activity as organic phosphates, adsorption to mineral surfaces, primarily Fe and Mn oxides, or by incorporation into phosphatic minerals such as Ca fluorapatite[4]. While many of these processes are abiotic, there is a clear and important role for biology through weathering and burial processes. In contrast, atmospheric $N_2$ is the largest surface reservoir of N, and its supply is virtually infinite. It enters biological cycles by microbial reduction to $NH_3$, and the majority of the fixed nitrogen in the oceans is returned to the atmosphere by microbial denitrification in the water column or in sediments[5]. $N_2$ is relatively inert, and its strong triple bond is broken during a reaction catalysed by the oxygen-sensitive enzyme nitrogenase[6]. Nitrogenase requires 38 atoms of iron per molecule of the protein megacomplex, which is the highest concentration of iron of any enzyme in nature[7,8]. Hence, $N_2$-fixation in the oceans may be controlled as much by soluble iron in the upper ocean as by an excess of P[9].

A long-debated question has been whether primary production is limited through the supply of N to the oceans, which is potentially controlled by iron limitation of nitrogen fixation, or if it is controlled by weathering and riverine fluxes of P[9–12]. The modern oceanic dissolved molar N:P ratio of 14.3 is lower than the planktonic N:P ratio of ~16.1, indicating N limits productivity on a global scale[1,13]. This is further cemented by the ability of $N_2$-fixers and phytoplankton to access dissolved organic P, which can exceed inorganic P by up to an order of magnitude in surface oligotrophic regions and represents a greater fraction of total dissolved P than that of dissolved organic N to total dissolved N (excluding $N_2$)[14–16].

While P is delivered by rivers, the Fe required by nitrogenase is primarily supplied to much of the open ocean by aeolian dust or volcanic ash[17,18]. Furthermore, since dust and volcanic events tend, at least in the modern ocean, to be sporadic, the flux of Fe can vary independently from that of P[19]. Over geological time, both the aeolian iron flux and weathering rates (P input) have changed considerably, and not

[1]Department of Earth and Planetary Sciences, Rutgers University, Piscataway, NJ, USA. [2]Department of Earth, Environmental and Planetary Sciences, Case Western Reserve University, Cleveland, OH, USA. [3]Department of Earth and Planetary Sciences and School of Engineering and Applied Sciences, Harvard University, Cambridge, MA, USA. [4]Laboratory for Climate and Ocean-Atmosphere Studies, Department of Atmospheric and Oceanic Sciences, School of Physics, Peking University, Beijing, China. [5]Institute of Ocean Research, Peking University, Beijing, China. [6]Department of Marine and Coastal Sciences, Rutgers University, New Brunswick, NJ, USA. ✉e-mail: linda.godfrey@rutgers.edu

necessarily in concert. The seawater inventory of P is dependent on fluxes into the ocean from global weathering and riverine fluxes and burial in sediments, but productivity depends on the supply of P and N to the euphotic zone, which is determined not just by the amount of N and P in the ocean, but also by vertical fluxes, especially in boundary upwelling zones driven by along-shore winds. It is in the oxygen minimum zones that N can be lost preferentially over P. Given these variables, can the balance between $N_2$-fixation and denitrification maintain a relatively stable N:P ratio or do other factors cause the N-cycle to uncouple from that of P?

The operation of the N cycle is clearly dependent upon global oceanic and orogenic processes, even as biological processes mediate between them. As with most biogeochemical cycles, much of our current understanding of the N cycle is based on the modern ocean. The palaeo-proxy of the N cycle is the record of N isotopes incorporated from dissolved inorganic N (DIN) into planktonic biomass, which is buried and retained in sediments, and reported relative to the atmosphere with differences of parts per thousand ($\delta^{15}N_{AIR} = (R_{sample}/R_{AIR} - 1) \times 1000$, where R is the $^{15}N/^{14}N$ ratio).

In the modern ocean, the value of $\delta^{15}N_{DIN}$ of the ocean nitrate reservoir is higher than the near 0‰ value of fixed N supplied through $N_2$-fixation due to the processes that remove fixed N. Localised incomplete denitrification that occurs in the OMZs within high productivity areas, where water that sank from the surface at high latitudes upwells along narrow shelves, leads to fixed N with high $\delta^{15}N$, whereas denitrification in sediments has a minimal effect on fixed N isotopes. In balancing the modern N isotope budget of the ocean, at least 70% of total denitrification has to occur in the sediments based on best estimates of water column denitrification, burial and source terms[20]. What then happens in terms of a balanced N cycle and the isotope composition of deep ocean nitrate if processes, such as water column denitrification and burial, have changed over time? Can we assume that the isotope composition of nitrate over geological time solely reflects relative losses of fixed N via denitrification in the water column relative to those in sediments?

To address changes in the operation of the geological N cycle, we present data from a series of cores deposited during the last 165 Myr and two models. The first is an isotope box model which addresses the impact on $\delta^{15}N$ of different rates of upwelling at a single site under two conditions: (1) in a setup closed for P without burial to represent biogeochemical cycling in a deep water column where P is completely remineralized, and (2) in a setup open for P that allows for exported material to be buried when productivity is dominated by shallow-water regions. The second model provides a set of fully coupled atmosphere-ocean general circulation model (AOGCM) simulations at different geological time slices with different continental configurations and $CO_2$ concentrations[21] which we use to estimate the total upwelling flux integrated over all coastal regions in 10 Myr time slices. The model uses Scotese continent configurations and global mean surface temperatures (GMST) reconstructed by Scotese and Wright[22] and Scotese et al.[23].

The marine N cycle is strongly influenced by $O_2$. If the $O_2$ supply is insufficient to maintain microbial aerobic remineralisation of organic matter (OM), nitrate is used as an alternative electron acceptor and N is lost from the ocean. However, this can give rise to complex feedback loops because the export of primary production, which drives microbial respiration at depth, depends not only on the total nutrient inventory, but on vertical mixing of deep nutrient-rich water to the surface. One self-regulating feedback between productivity and denitrification arises if productivity becomes very high and denitrification widespread, because the loss in DIN would lower productivity, and ultimately, denitrification would also decline. This leads to the assumption that it is unlikely that the ocean can ever be fully denitrified[24]. However, if N-fixing organisms were able to thrive in surface waters above areas of denitrification, they could

then hypothetically sustain a high productivity even in the case of a highly depleted oceanic N inventory. Thus, denitrification could continue to occur on a massive scale until the deep ocean had lost its entire fixed N inventory[25].

Oxygen is also important in defining the operation of sedimentary sinks of N and P. Within oxic systems, OM-bound N is released as $NH_4^+$, nitrified to nitrate and then escapes to the water column to fuel production or it can be used as a substrate for denitrification deeper in the sediment[26,27]. Sediments underlying anoxic water are more efficient at retaining N[28], but $NH_4^+$ released to the water column has two fates[29]. The first is nitrification higher in the water column with potential loss as $N_2$ at the anoxic-oxic interface by classical denitrification or anammox; the second is consumption.

Phosphate is removed from seawater and transported to sediments via the settling of OM and absorption on Fe-oxides. Most of the organic-bound P, like N, is remineralised aerobically within the water column, but P absorbed to Fe-oxides remains bound within sediments, locked in authigenic calcium fluorapatite (CFA) or transformed to refractory organic compounds through aerobic microbial activity[3,30,31]. Unlike N, the P burial efficiency in sediments overlain by oxic seawater is higher than in anoxic settings[3,32–35]. Under anoxic conditions, P is released during reductive dissolution of Fe(III)-oxides, if it is not first trapped by organic polyphosphates[36,37] or by ferrous phosphates that occur in coastal sediments before euxinic conditions are reached[38,39]. *Slomp and Van Cappellen*[37] suggested that when anoxic bottom water was more widely distributed, upwelling of P-enriched deep water increased productivity, but also redistributed P burial to the shelves in authigenic apatite, which is broadly confirmed by observation within the Benguela upwelling system[40]. The positive feedback of increased productivity, maintaining anoxic deepwater and inefficient P burial continues until, for whatever reason, productivity is curtailed by another nutrient, such as N, or upwelling ceases.

The way in which N and P respond to $O_2$ within the water column and within sediments makes it possible to decouple N and P and makes it possible to limit the amount of productivity that can be attained.

The sedimentary record of N isotopes provides a history of the N-cycle through time. The most common pathway whereby atmospheric nitrogen is fixed to form $NH_4^+$ has a minor isotopic fractionation: between − 2 and + 2 ‰[41]. In the presence of molecular oxygen, $NH_4^+$ can be oxidized by microbes to form nitrite and nitrate (i.e., nitrification). Where intense ocean upwelling creates high productivity, such as those in the eastern Pacific, along the Benguela or seasonally off the Arabian peninsula[42], hypoxic or anoxic conditions are created at mid-depths. As nitrate is the second most energic electron acceptor after molecular oxygen, it is used by heterotrophs during organic matter oxidation, ultimately leading to the formation of $N_2$. The reactions of denitrification, as well as nitrification, strongly discriminate between the two stable isotopes of N[43]. If these reactions are complete, which is generally the situation in the modern ocean for nitrification as well as sedimentary denitrification, then they leave no isotopic indicator of their occurrence in the sedimentary record. However, in the suite of denitrification reactions that occur in the water column, not all nitrate is consumed. $^{14}N$ is preferentially released to the atmosphere, and $^{15}N$ is retained as DIN, and the mixing of this isotopically enriched nitrate with the deep ocean nitrate reservoir imparts deep nitrate with its elevated $\delta^{15}N$ composition and the organic matter it supports[44–48].

Currently, $\delta^{15}N_{NO3}$ is + 5 ± 0.5‰[49], and organic matter that forms from this nitrate source has the same isotope composition. Continental shelves act as filters for bioavailable and organic N formed on land, with sediment denitrification removing more N than is delivered by rivers[50]. Deficits or excesses in $NO_3$ relative to $PO_4$, N* (defined as $NO_3^- -16 \times PO_4$) can be lower than −20 mmol N/m³ in water on the inner shelf. The deficit is made up by on-welling of nitrate from the deep ocean, which dominates fixed N fluxes to outer shelf-upper slope areas,

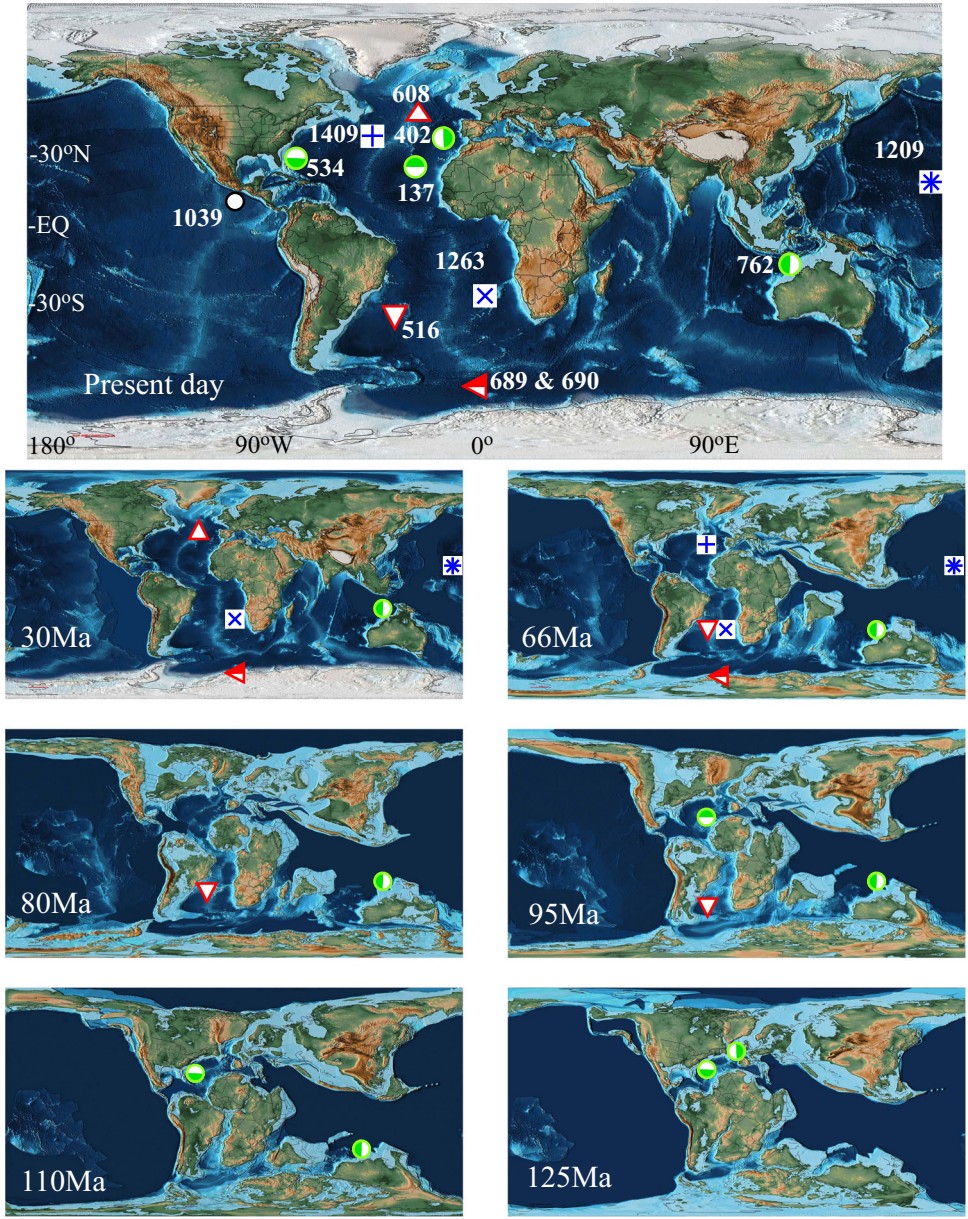

**Fig. 1 | Paleogeographic reconstructions showing the position of the DSDP and IODP holes used in this study.** PaleoDEM reconstructions of topography and bathymetry are taken from Scotese and Wright[22] and downloaded from EarthByte. The current location of each site is presented in the upper map. The paleo-reconstructions show the location of each site when the youngest sediment that was used in this study was deposited. The sites used to provide the $\delta^{15}N_{deep}$ record are indicated by green and white symbols, and for $\delta^{15}N_{distal}$, red and white symbols. The locations of the 3 holes used for foraminiferal bound $\delta^{15}N$ are included (blue and white symbols)[59].

indicated by higher N* values, reaching the positive values of +10 mmol N/m³ that typify the open North Atlantic[51,52]. With high rates of burial, organic N deposited at the outer shelf-upper slope of wide margins captures the $\delta^{15}N$ composition of deep nitrate[42,49], +5±0.5‰[42]. In contrast, sediments deposited along margins where upwelling produces intense productivity bear a high $\delta^{15}N$ signature, sometimes 10‰ higher than deep water nitrate. In distal areas of the open ocean, N that is deposited in sediments comes from the deep ocean via the thermocline, where nutrients accumulate by nutrient remineralization, eddy heaving and diffusion and reach the euphotic zone through horizontal diapycnal mixing, followed by isopycnal mixing, and finally through mesoscale eddies and wintertime convection[53]. The depth of the nutricline is important: if it is sufficiently shallow and nutrients are easily mixed into the euphotic zone, or if it is deep, and an important component of utilized N comes from $N_2$-fixation. Nutrient cycling can be more complex, generating $\delta^{15}N$ gradients in suspended and sinking organic N within the mixed layer related to nutrient recycling and fluxes into this layer[54,55]. In these areas, an important component of N comes from $N_2$-fixation, giving organic N in these areas lower $\delta^{15}N$ than deep nitrate[54,55].

## Results

The location of each core and its location when sediment accumulated is presented in Fig. 1; the N and C isotope data are presented in Fig. 2. Data for sea level and P accumulation rates are included in Fig. 2[4,56]. The $\delta^{15}N$ record is supplemented by published data[57–63].

### Sedimentary N isotopes since the middle Jurassic

Tethys was an established E-W oriented low-latitude ocean separating Gondwanaland to the south from Eurasia to the north when our $\delta^{15}N$ record starts[64,65]. Changes in continent configurations have altered both the position and fluxes of upwelling zones (Fig. 3). During the

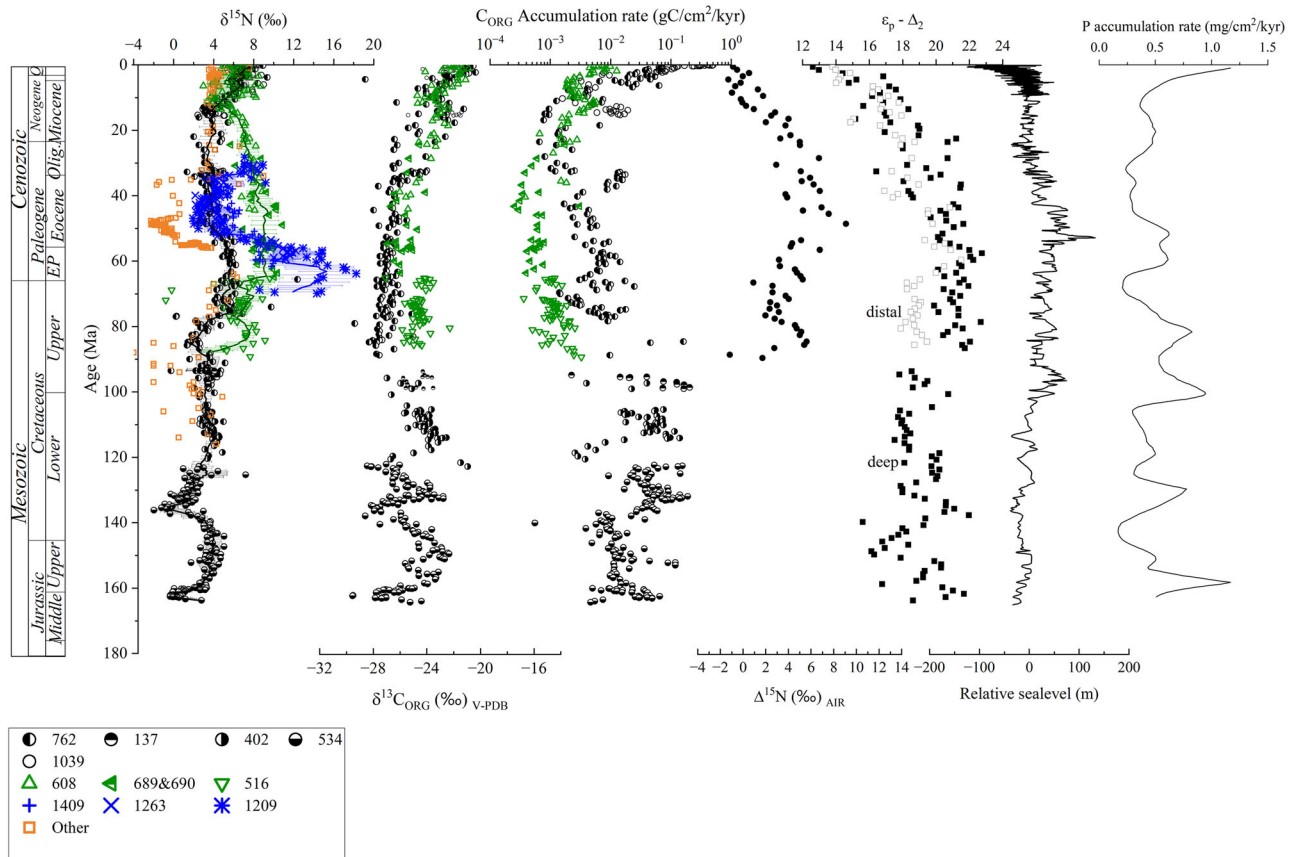

**Fig. 2 | Composite records of δ¹⁵N, δ¹³C_ORG, C_ORG accumulation rate, Δ¹⁵N, the difference between deep (black symbols) and distal (green symbols), δ¹⁵N, ε_p-Δ₂, which indicates the biological demand for CO₂ relative to available CO₂ (see text for its calculation).** Other data included are: sea level relative to today and P accumulation rate[4]. 762–516, bulk, this study; 1409–1209, foraminiferal bound[59]; Other: Meyers et al. [105], Sepulvida et al.[62], Li and Bebout[60], Robinson et al.[61], Clark[57], Knies[87], Liu er al.[46].

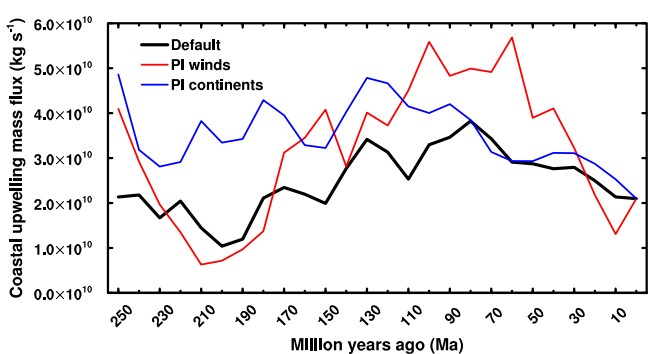

**Fig. 3 | Evolution of the globally spatially integrated coastal upwelling transport mass flux in the coupled climate model runs** comparing changing continent configurations and pre-industrial winds (red line) and changing winds and modern continent configuration (blue line).

Mesozoic, the long southern shore of Tethys provided the potential for high coastal upwelling fluxes of intermediate depth water that circulated within the tropics. Paleobathymetry suggests that deep and mid-depth water in the Tethys were likely formed within and confined to the Tethyan basins, closer to locations of upwelling than occurs today[66–68].

Multiple cores were used to create each record, and while the cores are from different oceans, there was no offset in δ¹⁵N when their records were spliced (Fig. 2). The temporal resolution of the core sampling does not allow us to capture any periodicity on orbital

frequencies, or geologically brief phenomena such as ocean anoxic events (Supplementary Fig. S1). Based on over 400 analyses, over the past 165 Myrs, δ¹⁵N_DEEP averaged 3.2 ± 1.6‰ (1 SD) while δ¹⁵N_DISTAL averaged 7.0 ± 1.8‰ (1 SD). Over this period, the δ¹⁵N_DEEP (deepwater nitrate) varied between −2 and +9‰, while the δ¹⁵N_DISTAL varied between +3 and +11‰ over the last 90 Myr (Fig. 1) (Supplementary Information). δ¹⁵N_DEEP and δ¹⁵N_DISTAL were the same at the start of the late Cretaceous and since the middle Miocene, while the rest of the time, δ¹⁵N_DISTAL has been higher than δ¹⁵N_DEEP, and by as much as 5‰ in the Paleocene.

Our data indicate that, with the exception of two relatively brief excursions to low values at 165 and again at 140 Ma, δ¹⁵N_DEEP has been fairly constant at 3.2 ± 1.6‰. Exceptions to this occurred 165 and 135 Ma when δ¹⁵N_DEEP was very low, and slightly elevated signals around the end Cretaceous to early Paleogene (75-55 Ma). The two negative excursions coincide with the opening of the Hispanic Corridor in the Western Tethys 165–160 Ma[69] and the Central American Seaway 140-130 Ma[67], which could have disrupted circulation patterns. δ¹⁵N_DISTAL has not been as uniform as δ¹⁵N_DEEP. During the latter half of the Cretaceous and first half of the Paleogene, δ¹⁵N_DEEP exceeded its 90 Myr average by as much as 4‰. Since we only have data from one core before 100 Ma, the comparison of the difference between δ¹⁵N_DISTAL and δ¹⁵N_DEEP, Δ¹⁵N, is limited to the last 100 Myrs. Δ¹⁵N was highest in the early Paleogene, indicating weak vertical mixing of deep nitrate to the surface ocean in the gyres. During the Miocene, Δ¹⁵N decreased to 0‰ even as both proxies increased abruptly (Fig. 2). This change indicates that vertical mixing increased during the Cenozoic. N isotopes in material bound in foraminifera (δ¹⁵N_FB) are within 1‰ of

$\delta^{15}N_{DISTAL}$ except for an excursion lasting less than 10 Myr around 60 Ma in the centre of the Pacific[59]. $\delta^{15}N_{FB}$ captures the $\delta^{15}N$ signature of nitrate below the mixed layer, but tends to neglect contributions from $N_2$-fixation. Thus, the extremely high $\delta^{15}N_{FB}$ relative to $\delta^{15}N_{DISTAL}$ indicates a highly stable water-column structure in the Panthalassa Ocean at that time, with little vertical mixing[59] (Fig. 1). Similarly to $\delta^{15}N_{DISTAL}$ and $\delta^{15}N_{DEEP}$, $\delta^{15}N_{FB}$ also abruptly increased around 10-8 Ma[63].

Organic carbon isotope data from the Exmouth Plateau (ODP site 763) off NW Australia overlap with those from open ocean sites[70], except for site 516 on the Rio Grande Rise, which is higher by ~ 2‰ in $\delta^{13}C_{ORG}$. From $\delta^{13}C_{ORG}$, we derived $\varepsilon_P$-$\Delta_2$, the term that reflects the demand for available C, and can be linked to productivity[71]. The maximal isotope fractionation associated with $CO_2$ uptake is reduced when C demand is high relative to available $CO_2$, driving down $\varepsilon_P$-$\Delta_2$. Changes in $\varepsilon_P$-$\Delta_2$ are largely the inverse of $\delta^{13}C_{ORG}$, with the highest values occurring at the end of the Cretaceous into the Paleogene, and the lowest in the last 10 Myr (Fig. 2). There is a slightly greater range in $\varepsilon_P$-$\Delta_2$ (9‰) compared to that of $\delta^{13}C_{ORG}$ (7‰) due to changes in $\delta^{13}C_{CARB}$[72]. Changes in $\Delta^{15}N$ and $\varepsilon_P$-$\Delta_2$ are similar (Fig. 2), which indicates that the contrast between $\delta^{15}N$ of OM produced in the gyres and margins (from deepwater nitrate) was greatest when C demand versus C availability was lowest, occurring around the Paleocene-Eocene boundary when Earth's climate was particularly warm[73].

Organic C accumulation rates were calculated using measured TOC concentrations and reported age models[72,74]. Accumulation rates of $C_{ORG}$ in margin sites during the latter half of the Jurassic and Cretaceous were between 0.003 and 0.20 g C/kyr/cm² and decreased over the Paleogene to a minimum of 0.001 g C/kyr/cm² during the middle Miocene, at which point they began to increase rapidly. $C_{ORG}$ accumulation rates in the distal areas from which $\delta^{15}N_{DISTAL}$ is derived were lower during most of the Cenozoic than at margin sites with a broad minimum throughout the Paleogene. Around 20 Ma, they increased to converge with the margin site values at the start of the Neogene (Fig. 1). Low $C_{ORG}$ accumulation rates correspond to times when $\delta^{13}C_{ORG}$ is low ($\varepsilon_P$-$\Delta_2$ high).

## Coastal upwelling and nitrogen isotopes

Today, the greatest losses of N in the water column, and where $\delta^{15}N$ of the ocean is defined, occur in the cold eastern boundary upwelling zones along the west coast of the Americas, Africa and the northern Indian Ocean[75]. Coastal upwelling occurs when the wind blows parallel to the coast, with the ocean to the right-hand side of the wind direction in the Northern hemisphere and to the left-hand side of the wind direction in the Southern hemisphere. In such configurations, the near-surface Ekman transport is offshore, forcing deep water to rise and replace surface water. Thus, coastal upwelling depends on both continental configuration (coastline orientation and location) and wind patterns. Upwelling can therefore change when the positions of the continents have shifted throughout geologic history, and when the wind patterns change due to climate warming or cooling over geologic time.

The climate model run at different time slices using the Scotese continent configuration[22] suggests upwelling may have varied significantly over the past hundreds of millions of years, maybe double modern values over much of the Cretaceous (refer to Supplementary information for global distributions of coastal upwelling mass fluxes for each of the 26 simulations). The calculation of the total coastal upwelling flux is described in the supplementary material and shown in Fig. 3 and the upper panel of S2. The results suggest that the upwelling flux can indeed change by factors of 5 over different geological periods due to the realigning of wind patterns and coastal regions, as depicted in the default simulation. We repeated the calculation twice more, first by varying the continental configuration using the actual wind pattern at each geologic time slice, and second by fixing the winds to pre-industrial values and using the continental configuration from each

time slice. These sensitivity tests indicate that continental configuration has the most impact on upwelling fluxes between 250 and 100 Ma. During this time period, the simulation with the preindustrial wind field combined with changing continent configuration approximately reproduces the global upwelling in the default simulation that includes changes in both the wind field and the continental configuration. In contrast, the simulation that used a modern continent configuration and wind fields derived from the different time slices does not reproduce the variability in upwelling generated by the default simulation between 250 and 100 Ma. Over the Cenozoic, the simulation using the modern continent configuration and changing wind fields reproduces the global upwelling in the default simulation better than the simulation with preindustrial winds and shifting continental configuration. It is important to note that coastal upwelling occurs in narrow bands near the coast at a much smaller scale than the climate model's resolution. Nonetheless, the calculation shows how upwelling responds to changes in both climate and continent configuration.

To first order, the nitrogen isotopic composition of marine sediments reflects the rate of denitrification in the water column, which is a function of $O_2$ concentration relative to its demand by heterotrophs as they remineralize OM[76]. For the model, $O_2$ concentrations are determined by $O_2$ solubility in surface water. As we kept the atmosphere's $O_2$ concentration and sea surface temperature (SST) constant, the only mechanisms that change subsurface $O_2$ concentrations are physical transport and aerobic respiration[70]. Transport also delivers nutrients to the surface water, which fuels productivity, and the consumption of productivity is the largest sink of $O_2$.

Intense upwelling creates OMZs, which forces denitrification and the remaining nitrate to acquire a high $\delta^{15}N$ signature. While residual nitrate is found at the surface above OMZs, its concentration is both low relative to that at the OMZ depth and much less enriched in $^{15}N$[77]. The $^{15}N$-enriched nitrate mixes into the deep-water inventory, which means that changes in upwelling rate are transmitted to deep water $\delta^{15}N$. This is what we aim to capture from near-margin sites which do not themselves host OMZs, rather than measuring $\delta^{15}N$ at a site which hosted an OMZ, which may be subject to site-specific characteristics of upwelling and productivity intensity. Today, upwelling occurs adjacent to narrow N-S orientated margins. Our model shows that changes in continent configuration have a profound effect on upwelling (Fig. 3) and that these areas ran E-W adjacent to wide shelves of Tethys (Fig. 1). Consequently, it is important to understand the response of the N-cycle not only to increased upwelling and productivity, but to increased burial of that productivity. The final set of models addresses the response of the N cycle to a change in the P inventory.

To investigate the impact of upwelling on $\delta^{15}N$, we varied the upwelling rate in the box model. The model (described in detail in the Supplemental Material S1) resolves an intermediate ocean box, in which the $O_2$ minimum develops, the overlying surface box, and a box representing the rest of the ocean (both surface and deep ocean). The surface and deep parts of the rest of the ocean are combined because the isotope composition of exported production is the same as the deep ocean. Since upwelling rates dictate productivity and the severity of the $O_2$ drawdown and increase in $\delta^{15}N$, we analyse the export and $\delta^{15}N$ associated with the 2 boxes representing upwelling zones.

We simulated upwelling rates between 0 and 0.25 Sv, which are typical values for one individual upwelling region. We used a model setup in which P is in a closed system to represent deep-water regions where P burial is relatively low (Fig. 4, panels A and C). To represent sites which are sufficiently shallow to allow significant burial of P, we subsequently modeled a system open to P (Fig. 4, panels B and D). In this setup, there is a P source due to weathering and a P sink through burial, consistent with our understanding of the natural P cycle[78]. The $\delta^{15}N$ of the surface ocean is 5‰ higher than the export production, which reflects the fractionation associated with biological utilization of nitrate.

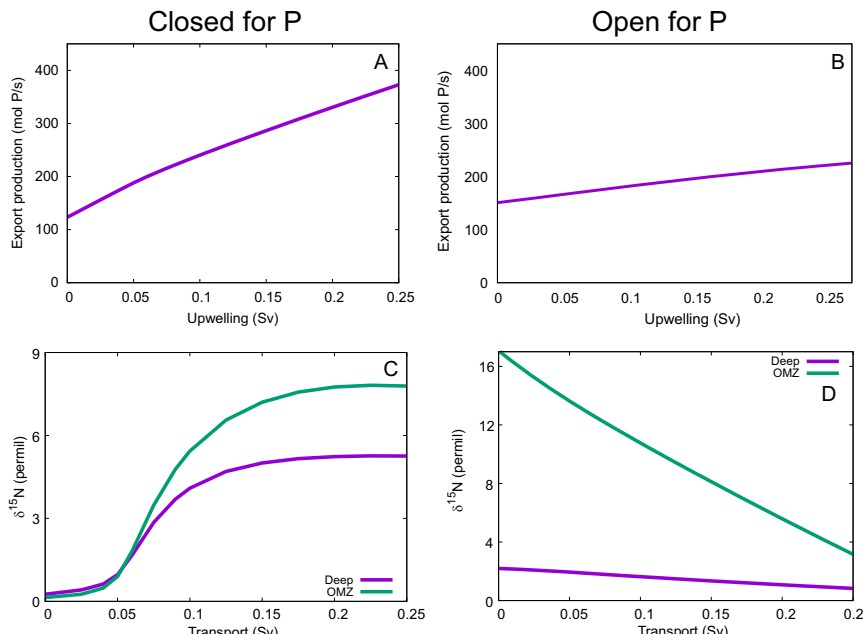

**Fig. 4 | Box model simulation results.** The relationship between upwelling transport and export production is qualitatively the same for the model setup closed for P (**A**) and for the open setup with weathering input and burial output of P (**B**). By contrast, the relationship between upwelling transport and $\delta^{15}N$ in both the deep ocean and the OMZ is qualitatively different for the closed-P setup (**C**), compared to the open-P setup (**D**). In the closed-P simulations, the P inventory is kept fixed at 2.42 Pmol. The P input is kept fixed at 2 mol/s in the open-P simulations.

In both the closed- and open-P systems, increasing upwelling transport leads to increased export production (Fig. 4a, b). In the closed-P system, $\delta^{15}N$ generally increases with increasing upwelling (Fig. 4c), since the enhanced export production leads to lower oxygen concentrations in the OMZ. In the open-P system, the increased export production leads to enhanced burial of organic matter, and a steady state is reached with a lower nutrient inventory. As a result, the oxygen transport and hence its concentration in the OMZ increases and $\delta^{15}N$ decreases (Fig. 4d). This is because (i) the $\delta^{15}N$ of DIN in both the deep ocean and the OMZ is determined by denitrification occurring in the OMZ, which makes the deep ocean entirely dependent on the OMZ in terms of its $\delta^{15}N$; (ii) almost all the exported N is supplied by upwelling from the deep ocean and almost all the upwelled N is exported. In the simulations for Fig. 4b, d, we assumed $\alpha = 0.25$, i.e., 25% of the exported organic P that is not remineralized either aerobically or through denitrification is buried. To investigate the sensitivity of the trend in Fig. 4d to model parameters, we varied the burial fraction $\alpha$ and the N:P ratio of burial. Furthermore, we included an increasing P burial efficiency with increasing oxygen concentration, a feedback that has been hypothesized to have played a role in Ocean Anoxic Events[79]. Although these changes affected $\delta^{15}N$ quantitatively, the trend of decreasing $\delta^{15}N$ with increasing upwelling remained. Thus, this qualitative result appears to depend solely on whether there is significant burial of organic matter. For shallow sites where organic matter burial is important, our model results suggest that increasing upwelling tends to increase $\delta^{15}N$. By contrast, our model results suggest that increasing upwelling in deep-sea regions tends to decrease $\delta^{15}N$.

Unsurprisingly, in the closed-P system, export production is determined by the P inventory (Fig. 5a). More specifically, export production increases with increasing P inventory up to about 2.3 Pmol (for comparison, the P inventory of the current ocean is ~ 3 Pmol) due to increasing P availability. Toward higher P inventories, the export production decreases again due to N limitation. In the open-P model setup, export production is essentially independent of the P input (Fig. 5b), as the higher input is compensated for by enhanced burial. There is a strong relationship between $\delta^{15}N$ and export production in the closed-P setup (Fig. 5c) but not in the open-P setup (Fig. 5d). Although export is constant, $\delta^{15}N$ decreases as a function of P input due to the increasing burial.

Overall, $\delta^{15}N$ behaves rather differently in the open P setup compared to the closed P setup. In the closed P condition, $\delta^{15}N$ is determined primarily by export production through its impact on oxygen concentrations in the OMZ and associated water-column denitrification. Export tends to decrease oxygen concentrations also in the open P setup. However, enhanced export leads to enhanced burial in this setup, both directly and as a result of decreased remineralization due to the lower oxygen contractions. The enhanced burial then decreases the nutrient inventory, which eventually leads to an increase in the oxygen concentrations in the OMZ. This in turn decreases water-column denitrification, which then leads to lower $\delta^{15}N$ values. Furthermore, the N burial itself tends to decrease $\delta^{15}N$. Although primary producers preferentially take up the lighter N isotope, there is essentially no fractionation due to export in the model because almost all the N in the surface box is consumed. As a result, exported N has approximately the same isotopic composition as the ocean average, which is isotopically heavier than N lost to water-column denitrification. Hence, the more N is lost due to burial (compared to water-column denitrification), the lower the oceanic N inventory's $\delta^{15}N$ becomes. Figure 6 highlights the essential differences between the closed (A) and open (B) models, as well as the differences between $\delta^{15}N$ in total OM and $\delta^{15}N_{FB}$.

## Discussion
### New insights into the N cycle
Our results strongly suggest that throughout the last 165 million years, the N cycle for the ocean as a whole has been remarkably stable. As recorded by deepwater $\delta^{15}N$ from near-margin sediments, less than 4% of the data fall outside the 95% confidence interval of + 2.8‰ from the median value of + 3.7‰. In contrast, $\delta^{15}N_{DISTAL}$ has been more variable. We have determined that the isotopic composition of fixed N (nitrate or ammonium) and resulting biomass is sensitive to both productivity and the consumption of $O_2$ when exported productivity is

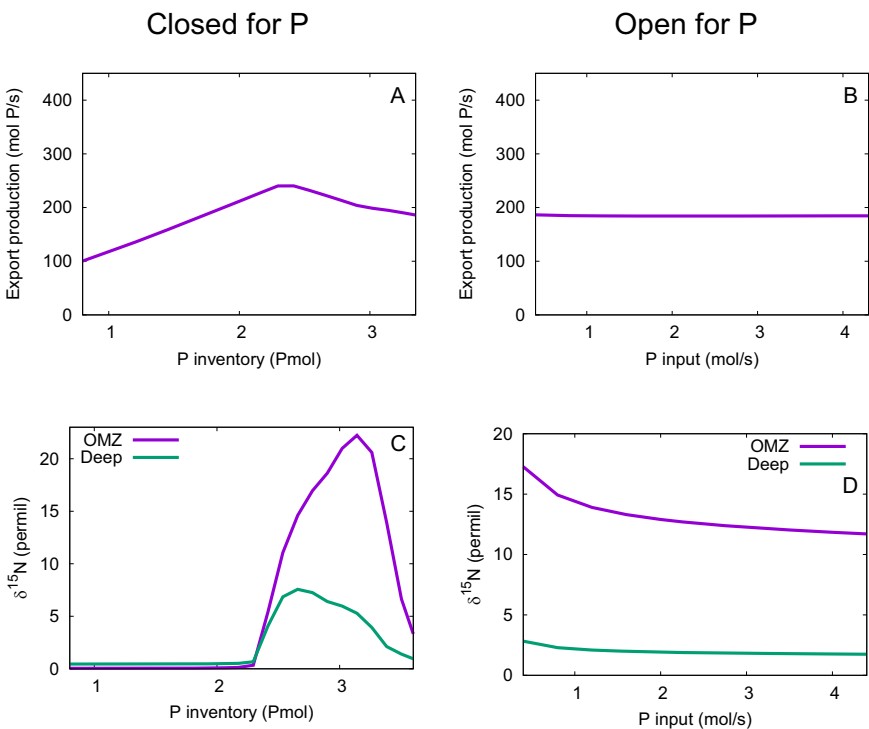

**Fig. 5 | Simulation results.** As the P inventory increases in the closed-P setup, export production first increases due to the increased P availability and then decreases as a result of N limitation (**A**). Export production is essentially independent of P input in the open-P setup (weathering input and burial output) (**B**), since the enhanced P input is compensated by enhanced burial. In the closed-P setup, the maximum in export production at an intermediate P inventory is approximately matched by maxima in $\delta^{15}N$ in the OMZ and deep ocean (**C**). There is a slight decrease in $\delta^{15}N$ as a function of P input in the open-P setup (**D**). The upwelling is kept fixed at 0.1 Sv in all the simulations.

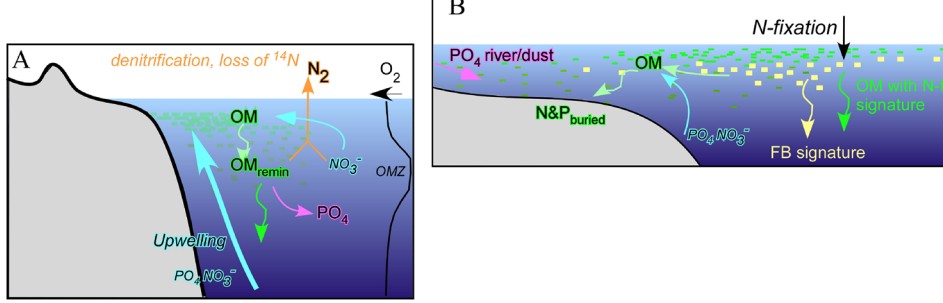

**Fig. 6 | Conceptual models for the cycling of N under closed and open conditions.** Schematic highlighting the features of the closed system (**A**) whereby nutrients (N and P) are returned to the water column, but in addition, denitrification occurs due to strong upwelling, and OM is imparted with a high causes minimal disturbance$^{15}$N. In (**B**), the open system, N and P are buried on shelves, and P is resupplied externally and N by $N_2$-fixation. Curved arrows indicate dissolved fluxes, wiggly arrows particulate fluxes.

remineralized. This is affected by circulation (vertical mixing) and the availability of other nutrients, but also by the role of $N_2$-fixers, which can 'short-circuit' the feedback associated with nitrate loss due to high productivity. Whether the associated nutrients return to the water column or are buried in sediments determines the impact on $\delta^{15}N$ of productivity changes due to vertical mixing. When nutrients are returned to the water column, increases in upwelling increase $\delta^{15}N$; the opposite happens when nutrients are buried.

In the box model, the rate of $N_2$-fixation is coupled to the deficit of N vs P for primary production. However, there exist regions in today's ocean where significant excess P occurs in surface water[80,81], which suggests a different control on $N_2$-fixation. Nitrogenase, nitrate and nitrite reductase enzymes all use Fe-rich cofactors. As with the N cycle, the marine cycle of Fe is regulated by redox potential. Fe is insoluble in oxic water, resulting in a very low, sub-1 nmol/kg concentration[82].

Without an external supply, such as dust or ash, Fe limitation can lead to large regions of the ocean with high nutrients but low chlorophyll (HNLC)[9]. Most of the P supplied to the ocean is via rivers, whereas Fe is supplied via dust or volcanic ash. Although the cycling of P and Fe in the ocean is affected by redox state, it is more important for Fe than for P. When bottom water is low in $O_2$, Fe undergoes reverse scavenging and is released from sediments, but when bottom water is oxic, Fe remains insoluble. Since we can account for the relationship between $\delta^{15}N$, $C_{ORG}$ accumulation and $\delta^{13}C_{ORG}$ over much of the last 165 Myr from our data without invoking severe N-limitation, adequate Fe must have been available. However, conditions in the early Cenozoic may have produced low dust fluxes to the central area of the Pacific Ocean, only punctuated by influxes of ash[83,84]. If low dust fluxes imposed strong Fe limitation on $N_2$-fixation, and the thermocline was deep, the very high $\delta^{15}N_{FB}$ in the most distal part of the Pacific may have been the

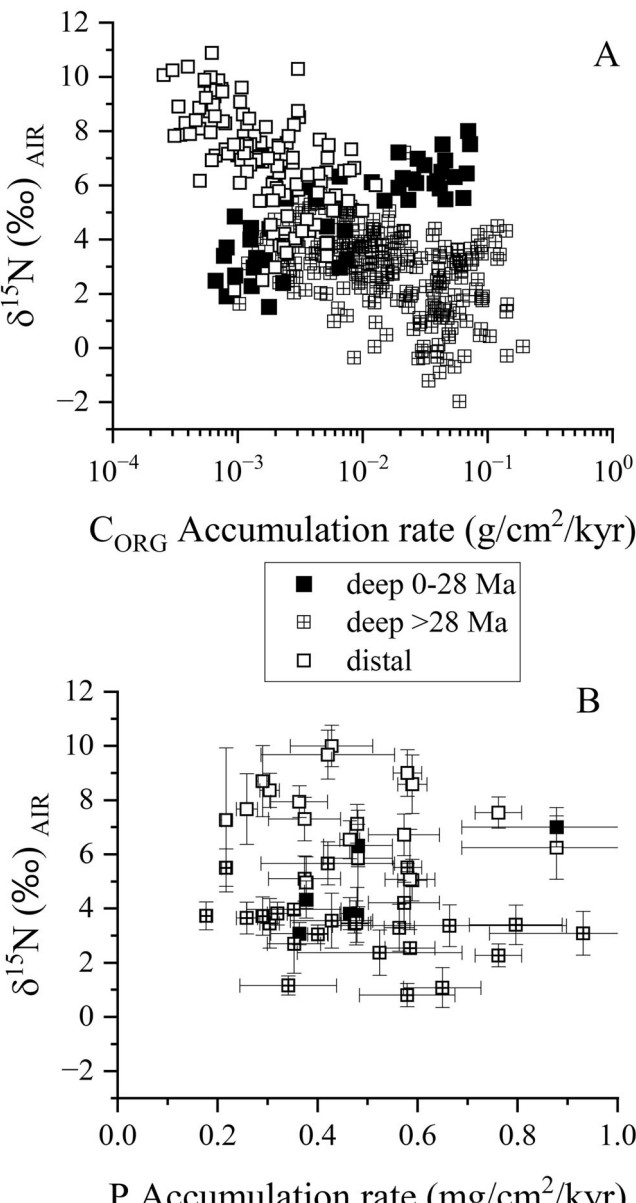

**Fig. 7 | N isotope response to changes in C and P accumulation rates.** The relationship between $\delta^{15}N$ and $C_{ORG}$ accumulation (**A**) shows a negative correlation for most of the record except for the last 28 Myr; while there is no relationship, with a possible exception for the last 28 Myr, with P accumulation rates (**B**). Data for (**B**) were binned in 5 Myr increments to allow for comparison with the P data from Follmi ([4]). Error bars are 1 S.D.

denitrification and burial as mechanisms for changing ocean $\delta^{15}N$. Therefore, the reversed trend in the open-P model has to be caused by burial. Furthermore, the degree to which ocean P is conserved within the ocean (closed) varies between very conserved, meaning P has a long ocean residence time, and minimally conserved, meaning a short residence time. When P is highly conserved, the model indicates one key mechanism that changes ocean $\delta^{15}N$: water-column denitrification. As P becomes more dynamic, i.e., when the inputs or outputs increase relative to the inventory, then both water-column denitrification and burial will change ocean $\delta^{15}N$. Under these conditions, the trend of decreasing $\delta^{15}N$ when upwelling fluxes increase in the open-P setup must be caused by burial (Fig. 4).

A factor of 2 to 3 change in upwelling over the studied time is predicted by the climate model (Fig. 4), which both the closed and open P setups show will lead to measurable changes in productivity and N isotopes. While an increase in upwelling always increases productivity, $\delta^{15}N$ increases in the static model but decreases in the open setup. Relative to the closed setup, the upwelling-driven changes in $\delta^{15}N$ in the open setup are more subdued for deep water, but more pronounced in areas of OMZs over the same change in upwelling (Fig. 4C and D). Comparing $\delta^{15}N$ to indicators of productivity will resolve the model that is most dominant at any one time if upwelling is known to change.

Determining which model better describes the $\delta^{15}N$ record regarding N's response to variations in the ocean's P inventory is less clear. When burial is included, there is almost no effect on export (because the model requires the P inventory to be in steady state) or $\delta^{15}N$. When P is conserved, there is an increase in export production and $\delta^{15}N$ with increasing P, until a threshold is reached, and N becomes limiting. Around this point, $\delta^{15}N$ in the deep ocean and OMZ attains its maximum value. There are no good proxies for the P inventory over geological time, and due to the role of the bottom water redox state, which is affected by export production, a feedback between the amount of P in seawater and upwelling exists. Nevertheless, given these constraints, we can determine which model best describes a period in time, i.e., when P is conserved or not, and why.

Deep-water $\delta^{15}N$ plotted against $C_{ORG}$ accumulation rate shows two responses (Fig. 7a). Except for the last 30 Myr, and possibly between 108 to 113 Myr, there is a negative correlation between these parameters (Fig. 7a). In contrast, there is almost no correlation between deep water $\delta^{15}N$ and P accumulation, with the possible caveat of a positive correlation from 30 Myr to present (Fig. 7b). Given these relationships, the data indicate that only in the Neogene and maybe around the K-Pg boundary has there been greater conservation of P in the water column while burial has, for much of the last 165 Myrs, been of greater importance.

$\delta^{15}N$ in distal areas of the ocean also show a decrease in value as $C_{ORG}$ accumulation rates increase (Fig. 7a), extending the trend defined by samples representing deep water environments. The trend between $\delta^{15}N_{DISTAL}$ and $C_{ORG}$ is at slightly higher $\delta^{15}N$ than the trend between $\delta^{15}N_{DEEP}$ and $C_{ORG}$ which is most likely related to preservation, as oxic diagenesis has been shown to increase in $\delta^{15}N$ by about $1.0-.5‰$ with a 50% reduction in $N_{ORG}$[85]. While diagenesis must have elevated the $\delta^{15}N$ signal in these carbonate-rich, low-OM sediments, the similarities in the temporal pattern to that of $\delta^{15}N_{FB}$, which is even higher[59], indicate that the information $\delta^{15}N_{DISTAL}$ carries is robust. Unlike the deep-water trend, samples that form the lower $\delta^{15}N$ end of the distal trend were deposited during the Neogene, and overlap with the deep-water trend, setting $\Delta^{15}N$ equal to 0 (Fig. 2). As with deep water, we find no correlation with P accumulation (Fig. 7b).

These observations lead to a fundamental conclusion that the closed-P model describes the N system since the Neogene, but is not representative of the majority of the last 165 Myr

We can start to infer the causes behind these relationships from a closer look at the data, the model and paleogeography. For much of

result, and at these times, low productivity resulted from severe N limitation.

While P can be more or less conserved at different time periods and ocean basins, there will always be some input and output of P. In other words, a model setup open for P will always be a more complete representation of the oceanic P cycle than one with a closed ocean P inventory, regardless of the paleo-record or paleo-environment. The comparison between the setups with closed and open P cycles serves to disentangle causes and effects, in addition to illuminating the different impacts that upwelling may have in coastal and open-ocean environments. In particular, the closed-P model includes only one key mechanism for changing ocean $\delta^{15}N$: water-column denitrification. Thus, the closed-P simulations allowed us to focus on this mechanism in isolation. The open-P model includes both water-column

the Mesozoic portion of the record, the data are from the western section of Tethys which was characterized by long coastlines and broad shelves even though Hole 534 was drilled in deep water (Fig. 1). Run-off and high rates of upwelling combined to lead to robust productivity (Fig. 2) yand burial. The middle Late Cretaceous marks the start of convergence between Africa and Eurasia starting near Arabia and the Eastern Mediterranean, when passive margins became fore-deep settings[64]. As fault-bound basins deepened and platforms were reduced in size or lost, upwelling could have occurred over increasingly deep water, increasing water-column remineralisation and recycling of N and P. The accumulation of $C_{ORG}$ began to decline at the start of the Late Cretaceous and had become very low in the ocean centers by the Santonian at 86 Ma. While the configuration of the continents allowed coastal upwelling to remain high well into the Cenozoic, the large difference between $\delta^{15}N_{DEEP}$ and $\delta^{15}N_{DISTAL}$ suggests that vertical mixing was low (open-system P) at the sites where our samples were deposited, or there was a very large increase in the P inventory (closed-system P). Given that the climate model suggests increased vertical mixing, the suggestion that the extremely high $\delta^{15}N_{FB}$ signature originated in another area of the ocean[59] and was exported in surface waters to distal parts of the ocean implies that parts of the ocean may follow different nutrient models. Whether this situation might have occurred at the fault-bound blocks of what became the eastern Mediterranean or close to the India-Asia collision zone is unknown. Alternatively, another nutrient may have limited the amount of $N_2$-fixation so that high $\delta^{15}N$ signatures were not minimised in other parts of the ocean. Low dust fluxes to remote areas of the Pacific throughout much of the late Cretaceous until the beginning of the Miocene[83,84,86] suggest that other nutrient to be Fe.

A minimum in $\delta^{15}N_{DEEP}$ occurs during the Eocene, which is recorded in $\delta^{15}N_{FB}$[59] and is particularly pronounced in bulk $\delta^{15}N$ from the Arctic[87]. The Arctic $\delta^{15}N$ minimum was attributed to increased $N_2$-fixation in a restricted basin, and its increase to circulation changes and inflow of Atlantic water. Changes in circulation were also proposed to cause the $\delta^{15}N$ changes recorded in the eastern equatorial Pacific[61], related to isolation of Antarctica[88], and were integral to how the N cycle responded towards open or closed nutrient dynamics.

Towards the start of the Neogene, the relationship between $C_{ORG}$ accumulation and $\delta^{15}N$ changed. It became consistent with the closed P model, indicating the importance of nutrient recycling relative to burial. Tethys was no longer an unrestricted circum-Equatorial ocean, and as the climate cooled, wind fields and overturning circulation increased. Wind-driven coastal upwelling is recorded off Peru in the early Miocene but became widespread along the South American Pacific coast by the middle to late Miocene[89,90]. $\delta^{15}N_{DEEP}$ had started to increase around 28 Ma, while $\delta^{15}N_{DISTAL}$ declined. These parameters converged around 11 Ma, demonstrating the importance of diffuse upwelling of deep water in even the most distal areas of the ocean[91].

The configuration of the continents suggests that wind-driven coastal upwelling at the end of the Cenozoic was not as vigorous as it had been when it occurred along the coastline of Tethys (Fig. 3). However, the late Cenozoic upwelling systems occurred predominantly over deep water, allowing more export production to be remineralised rather than buried, and under this scenario, $\delta^{15}N$ increases. Upwelled water was also colder than before, having downwelled at higher latitudes, making areas of intense upwelling less favourable to $N_2$-fixers compared to upwelling that occurred within Tethys. This modern-style overturning circulation also separated areas of high production and $N_2$-fixation[81], making attainable production in many areas heavily dependent upon N availability through circulation rather than local $N_2$-fixation. The last 28 Myr is also the only period when a correlation (positive) exists between P accumulation rates and $\delta^{15}N_{DEEP}$, indicating the role of a higher P inventory due to uplift and erosion. The increased delivery of nutrients to the surface water, P through runoff, and N through remineralization and vertical mixing, is

also reflected by the greater demand for $CO_2$ relative to available $CO_2$, as evidenced by low $\varepsilon_P$-$\Delta_2$.

In conclusion, evaluating records of $\delta^{15}N$ that represent nitrate in deep water and distal thermoclines in a framework provided by a set of coupled P-N isotope models, we find that the geometry, orientation and latitude of coastlines along which upwelling occurred in the geological past is key to understanding upwelling rates and its role on the N cycle. Large differences between $\delta^{15}N_{DEEP}$ and $\delta^{15}N_{DISTAL}$ (high $\Delta^{15}N$) indicate strong stratification if $C_{ORG}$ is also low, whereas an increase in the P inventory should be accompanied by greater accumulation of $C_{ORG}$. Conversely, low $\Delta^{15}N$ values suggest increased vertical mixing. While we assume that $N_2$-fixation has always been able to restore deficits in DIN, this is only possible with sufficient Fe. Our analysis suggests that Fe supplies might have been limited, and that $N_2$-fixation, especially in parts of the vast Panthalassic Ocean, was restricted, leading to extremely high $\delta^{15}N$. Conversely, there can be Fe limitation on nutrient uptake, which could result in unutilised nutrients (as is currently the case in the Southern Ocean). This in turn leads to high $\delta^{15}N$ of nitrate at the ocean surface and low $\delta^{15}N$ in exported organic matter. Increased meridional overturning circulation and the formation of strong frontal zones in the latter part of the Cenozoic has separated areas of $N_2$-fixation (low $\delta^{15}N$ at the surface) from upwelling and denitrification (high $\delta^{15}N$ in OMZs) today, which has made them clearly distinguishable.

# Methods
## Materials
Sediments from a series of cores, described in the online supplement, deposited during the last 165 Myr were used to give bulk sediment N isotopes and C isotopes in organic matter. Data from different sites provide N isotope data representing different aspects of the ocean N cycle. DIN incorporated into planktonic biomass is buried and retained in sediments, providing the geological record of $\delta^{15}N$[2,48,49]. With high rates of burial, organic N deposited at the outer shelf-upper slope of wide margins captures the $\delta^{15}N$ composition of deep nitrate[42,49] and can create a record of deepwater $\delta^{15}N$ ($\delta^{15}N_{DEEP}$)[42,49,53]. Because organisms residing in the photic zone may not acquire their N solely from the deep ocean, the geological record can differ from that of deepwater DIN, and we refer to it as $\delta^{15}N_{DISTAL}$. Our record of $\delta^{15}N_{DISTAL}$ is obtained from sediments deposited far from the margins.

We define the difference between $\delta^{15}N_{DEEP}$ and $\delta^{15}N_{DISTAL}$ as $\Delta^{15}N$. The value of $\Delta^{15}N$ reflects mixing of deep water into the surface in distal areas of the ocean, approaching 0 when the ocean is well mixed. We also include planktonic foraminifera-bound nitrogen isotopes, $\delta^{15}N_{FB}$, reported by Kast et al.[59]. $\delta^{15}N_{DISTAL}$ and $\delta^{15}N_{FB}$ should be similar, given the sample locations used for the $\delta^{15}N_{FB}$ record. However, analyses of particulate and bulk sediment $\delta^{15}N$ and $\delta^{15}N_{FB}$ do not always match. For example, in the Sargasso Sea, $\delta^{15}N_{FB}$ reflects the $\delta^{15}N$ of nitrate ($\delta^{15}N_{NO3}$) in the upper thermocline[92–94] (+ 3‰) which differs from deep water $\delta^{15}N_{NO3}$ (- 5‰)[49,54,92–94]. In the South China Sea, the range in $\delta^{15}N$ of both foraminifera-bound and all bulk sediment for all South China Sea cores are similar (+ 4 to + 8‰) over the last 40 kyrs, although the difference between $\delta^{15}N_{BULK}$ and $\delta^{15}N_{FB}$ for individual cores ranges from − 2 to + 3‰[95][58]. A large component of terrestrial N in the bulk record was thought to be the cause of the discrepancy, which is possible given the lack of change in $\delta^{15}N_{BULK}$ not typically seen in other locations, e.g.[95].

Organic C isotope ($\delta^{13}C_{ORG}$) data from a core retrieved from the Exmouth Plateau off NW Australia[74] supplement C isotope data reported previously[70,72]. Since $\delta^{13}C_{ORG}$ depends on the isotope composition of inorganic C, the value of $\delta^{13}C_{DIC}$ needs to be accounted for in order to address productivity. Hayes et al.[71] defined the relationship $\varepsilon_P$-$\Delta_2$ = $\varepsilon_{TOC}$ - $\Delta_{CARB}$, which can be calculated from our data: $\varepsilon_{TOC}$ is the difference between $\delta^{13}C_{CARB}$ and $\delta^{13}C_{ORG}$; and $\Delta_{CARB}$ is the isotope difference between dissolved $CO_2$ and carbonate minerals[72]. The

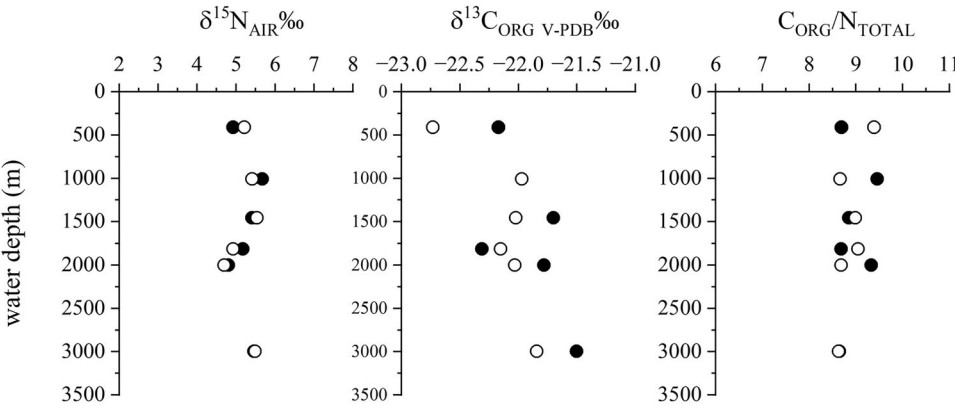

**Fig. 8** | N and $C_{ORG}$ isotope and $C_{ORG}/N_{TOTAL}$ data for sediments deposited off Cape Hatteras, NC, USA. Closed symbols 0-1 cm depth, open symbols 1-2 cm depth.

isotope composition of dissolved $CO_2$ was estimated using temperature estimates from Song et al.[96] and a $\delta^{13}C$ value of pre-industrial atmospheric $CO_2$ of -5‰, and equilibrium fractionation factors for hydration of $CO_2$ and then formation of $CaCO_3$ from Mook[97]. While the preindustrial value of $CO_2$ has likely changed, we assume it is responsible for a small component of the total change in sedimentary $\delta^{13}C$. $\Delta_2$ accounts for secondary biological processes that cause primary production to differ from sedimentary organic C, such as trophic enrichment of $^{13}C$ or its depletion by chemotrophy. For oxic systems, $\Delta_2$ is typically 1.5‰ and is not thought to have changed over Phanerozoic time[71]. Further description of this is given in Freeman and Hayes[98] and Hayes et al.[71]. The difference between the isotopic composition of organic and carbonate C was found to decrease after the early Oligocene in response to high growth rates relative to $CO_2$[71].

**Analytical methods**

Samples were homogenized and analyzed by CF-IRMS (Eurovector 3000 coupled to a GVI Isoprime 100). Samples were analyzed separately for $\delta^{15}N$ and $\delta^{13}C_{org}$. Samples were not pretreated for $\delta^{15}N_{SED}$, and a $CO_2$ trap was used to increase sample throughput due to a potentially long $CO_2$ bleed. For $\delta^{13}C_{org}$, carbonate was removed with 25% HCl and the residue washed with deionized water. All samples were loaded in tin capsules. Nitrogen and organic C concentrations were determined from the area of the major ion (28 or 44) area and calibrated with acetanilide. Isotope standards (IAEA N-1 and N-3 for $\delta^{15}N$, NIST 22 and IAEA CH6 for $\delta^{13}C$[99,100]) were included in every run, separating 8 samples, with an in-house sediment standard included, which was core-top material from a metalliferous carbonate-rich sediment from the eastern low latitude Pacific.

We evaluated the potential for inputs of terrestrial N using $C_{ORG}/N_{TOT}$ and $C_{ORG}$ accumulation rates using the following approach: Sediments deposited on the continental margins and perhaps in less than 3000 m water depth, as well, accumulate rapidly enough that OM quickly becomes isolated from the effects of oxic diagenesis on N isotopes that occurs on the seafloor[42,101]. Sediment in deep-water areas accumulates at about one-third the rate or less and has lower $C_{ORG}$, and they are more susceptible to diagenesis and changes in $\delta^{15}N_{SED}$[101,102]. While $\delta^{15}N$ of slowly accumulating sediments can be higher than sediment-trap or deep-water $NO_3^-$, this offset is considered to be constant, so that temporal variations are not a function of changing preservation at one location but reflect changes in the water column processes that determine initial $\delta^{15}N$[101].

Sedimentary $\delta^{15}N$ data from either margin sites or open ocean carbonate-rich sites seamlessly splice to give a record with no breaks as data switch between different cores. This indicates that diagenetic changes were minimal, or data from each core was similarly affected, even though the sites are distant from each other, but importantly, water column conditions overlying each site were the same at any one

time. At any one point in time, there may be as much as a 2‰ spread, which may be due to short (sub-millennial) variability which has been recorded throughout the late Pleistocene[46]. However, our intent is to address long-term changes where variations are larger, and this degree of variation is acceptable and consistent with other studies[42,101].

The $\delta^{15}N_{SED}$ average of 6.7‰ for our Pleistocene to recent samples is the same as the global average reported by Tesdal et al.[42], and about 1.5‰ higher than deep water nitrate. This offset may arise from a trophic increase in $\delta^{15}N$ of exported N relative to primary production[103], or diagenesis at the seafloor under oxic water[42,101,104]. In a series of experiments, Lehmann et al.[104] suggested that even under oxic conditions, $^{14}N$-enriched bacterial biomass could restore $\delta^{15}N$ to initial values following early increases in $\delta^{15}N$ relative to settling organic matter, while under low $O_2$ conditions, organic matter degradation led to minor, <1‰ increases in $\delta^{15}N$. Rather than explaining changes in $\delta^{15}N$ as a reflection of microbial-induced change within sediments, we assume ocean nitrate $\delta^{15}N$ changes over time.

A test was performed to determine whether margin sites distal to upwelling systems or major (i.e., Mississippi-size) rivers record deep-water $\delta^{15}N$ due to onlapping of mid-depth water onto the shelf and the inner shelf is effective at trapping terrestrial material. Core-top sediment from a multicore, which causes minimal disturbance to the sediment surface, deployment across the shelf and slope at Cape Hatteras (South Atlantic Bight) was obtained (Dr Yair Rosenthal, Rutgers U) and analysed for total $\delta^{15}N$ and organic $\delta^{13}C$ using the methods as described above. Water depths ranged from 410 m to 2997 m.

From each of the 6 sites, samples were taken at 0-1 cm and 1-2 cm intervals (Fig. 8, online data). The average $\delta^{15}N$ for 0-1 cm was +5.24±0.34‰ and for 1-2 cm + 5.21 ± 0.34‰. The average $\delta^{13}C_{ORG}$ for 0-1 cm was −21.92±0.30‰ and for 1-2 cm −22.13±0.31‰. The average $C_{ORG}/N_{TOTAL}$ for 0-1 cm was 8.94±0.36‰ and for 1-2 cm 8.90±0.30‰. There is no change in $\delta^{15}N$ with depth at the 95% confidence level, though there is a minimal drop in $\delta^{15}N$ for the samples at 2000 m, which corresponds to the maximum depth of the Gulf Stream (Andres, 2021). Depth related changes in $\delta^{13}C_{ORG}$ are slightly more pronounced, and are slightly lower at the 1-2 cm depth than the top 1 cm. The only sample exhibiting a notable difference is the shallowest 1-2 cm sample, with $\delta^{13}C_{ORG}$ less than −22.5‰, which could indicate a minor terrestrial C contribution.

## Data availability

The isotope data generated in this study are available via Zenodo at https://doi.org/10.5281/zenodo.16591434. Global distributions of coastal upwelling mass fluxes for each of the 26 simulations are available via Zenodo at https://doi.org/10.5281/zenodo.16828679.

## Code availability

The code of the multi-box box model of the nitrogen cycle (including technical instructions to compile and run the model) is available

through A.W. Omta's GitHub page: https://github.com/AWO-code/N-cycle. The source code of CESM1.2.2 can be accessed at https://www.cesm.ucar.edu/models/cesm1.2.

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

## Acknowledgements

We wish to thank Dr C. Seldon and the reviewers for their comments. This study was supported by NSF OCE 0084032 Biocomplexity: The Evolution and the Radiation of Eucaryotic Phytoplankton Taxa (EREUPT). Samples used in this study were provided by the Ocean Drilling Program (ODP), which is sponsored by the U.S. National Science Foundation (NSF) and countries participating under the management of the Joint Oceanographic Institutions (JOI), Inc. L.V.G. was supported by the Agouron Foundation. A.W.O. was supported by the Simons Collaboration on Computational Biogeochemical Modeling of Marine Ecosystems/CBIOMES (Grant ID: 549931, MJF). E.T. was funded by NSF grant 2303486 from the P4CLIMATE program and thanks the Weizmann Institute for its hospitality during parts of this work. Y.H. and X.L. are supported by the National Natural Science Foundation of China (NSFC) (Grant ID: 42488201). P.G.F. was funded by the Bennett L. Smith Endowment.

## Author contributions

L.V.G.: conceptualization, data collection, writing original draft, and editing; A.W.O.: writing box model code, performing simulations, and editing; P.G.F.: conceptualization and editing; Y.H. and X.L.: providing AOGCM simulation data; E.T., Y.H., and X.L.: analyzing AOGCM results.

## Competing interests

The authors declare no competing interests.
