## [Peer Review file · Nature Communications]

Stability of the marine nitrogen cycle over the past 165 million years

Corresponding Author: Dr Linda Godfrey

Version 0:

Reviewer comments:

Reviewer #1

(Remarks to the Author)

The authors present new nitrogen and carbon isotope data from marine sediments spanning from 165Ma to the modern and use these data, along with a computational box model, to assess linkages between coastal upwelling, biological productivity, and P and N cycling. The results reveal previously undocumented feedbacks between physical oceanographic conditions and biogeochemical cycling that help explain patterns in the data that were previously not well understood.

I have spent a lot of my time thinking about linkages between nitrogen isotopes, redox and productivity, and I am excited to see how these parameters can be linked to global oceanographic and orogenic processes. Many patterns in paleoenvironmental proxy data have always been difficult to explain by biogeochemistry alone, and so this linkage to physical and geological phenomena that operate on a global scale is refreshing. I could imagine that these findings will lead to new hypothesis about biogeochemical cycles even further back in time.

I do, however, have some comments that should be taken into account to strengthen the conclusions.

1) Despite the extensive effort to accumulate such a large dataset, I worry that the data are not globally representative, given limitations of availability. According to Fig. 1, several of the sites appear to be located in possibly restricted basins. In the modern ocean, restricted basins can be distinct in $\delta^{15}\text{N}$ from open marine margins. The supplementary file needs a more detailed description of the major sampling sites to resolve this concern.

2) I'm not entirely convinced by the assumption that $\delta^{15}\text{N}$ along coastal margins captures deep marine nitrate. Tesdal et al. 2013 (Biogeosciences) show that the highest $\delta^{15}\text{N}$ values in sediments tend to coincide with upwelling zones. Here, denitrification rates are locally high, causing dissolved nitrate to be locally enriched in $\delta^{15}\text{N}$ (up to +12permil, in fact consistent with some of the model results in Fig. 3), but this is likely not representative of the global deep marine nitrate pool. It would help if modern data could be used to illustrate the overall concept of comparing marginal to gyre sediments as a proxy for nitrate advection.

3) It makes a significant difference in the model results whether P is treated as static or dynamic, but it is not made clear how these two scenarios can be gleaned from the paleo-record. When discussing the model results later on, more information (ideally quantitative) is needed for how the state of P can be constrained.

Minor points:

II. 74-75: "We explore whether productivity followed P input to the surface ocean" – please explain on which timescales you are exploring this and by which means. This sentence actually seems to be out of place. Maybe delete it or move it further down where you transition to describing the aims of this study.

I. 78: Measured N:P ratios in what? I assume in seawater, but it could also be in biomass. Please clarify in this instance.

L. 121: Are $\delta^{15}\text{N}_{\text{GYRE}}$ and $\delta^{15}\text{N}_{\text{FB}}$ are similar when the foraminifera are obtained from gyre regions as well? Or do

foraminifera from any part of the ocean match $\delta^{15}\text{N}_{\text{GYRE}}$? Please clarify.

I. 123: "in the upper thermocline" – this presumably depends on the species of foraminifera and their preferred habitat, or not?

Fig 1 and associated text (II. 144-153): The text states that both marginal and gyre sites are shown in Fig. 1; however, the figure's legend lists margin sites, carbonate sites and foraminifera bound samples. Gyre samples are not mentioned in the legend. Please highlight those. Carbonate samples have not been mentioned in the text until now. It would be good to explain how those compare to the margin and gyre sites. Amendment: With further reading, I get the impression that the carbonate sites are the gyre sites. That confuses me, because a lot of carbonate precipitates in shallow waters. So some clarification is needed.

Fig. 1 C-accumulation: Change the axis labels into a more readable notation instead of Excel's version (e.g. 10^{-4} instead of $1.E+04$). Also, should the units not have a surface area in them? (e.g., g/kyr/m^2 or something like that?). If this is a global C burial rate, then the values seem very small.

II. 236-237: something is wrong with this sentence. Please rewrite.

I. 361: demand for CO_2 instead of demand for C?

II. 304-399: This review section seems a bit out of place. It made sense to have the modelling after the presentation of the data, but after reading about the model, I expected to see the model results linked to the data presented above. But that's not the case. Instead, the manuscript continues with a more general review of other topics, and the linkage between the data and the model does not occur until I. 400 onwards. I found that confusing. Please consider reorganising the text. The review portion between II. 304-399 could probably be shortened and perhaps move up or merged with later parts of the discussion.

L. 423: How does one know if sediments play an important role in nutrient storage? This requires explanation.

II. 448-450: This sentence seems to contradict the previous one, according to which upwelling over deep water caused efficient remineralization of OM before settling on the seafloor (and thus presumably low OM accumulation). If that was the case, then the P inventory must not necessarily have been lower to explain low OM accumulation.

I. 469: It would be good if quantitative temperature thresholds could be provided for N-fixers and for the inferred water conditions.

Methods: State the reproducibility of samples and quality control standards

Which software was used to set up the model? It would be ideal if the code could be shared in the appendix.

Please share an Excel spreadsheet with all the new sedimentary N and C data.

Eva Stüeken

Reviewer #2

(Remarks to the Author)

This study aims to analyse the relationship between the nitrogen and carbon cycles over the last 165 Myrs. It presents new nitrogen and carbon isotope data from a series of cores. It combines it with a global climate model to reconstruct ocean upwelling and a box model to characterise the dynamics between the nitrogen isotopes, upwelling and production. This study presents some interesting and potentially important results, including the role of organic matter burial, upwelling and iron on the nitrogen isotope values and the implications for long-term nitrogen and carbon cycles. However, the paper covers many aspects that need to be better connected, making it challenging to identify the key findings.

The paper has the potential to have significant results for Nature Communications. It presents the first long-term record of nitrogen and carbon isotopes and combines modelling to examine the controlling processes. However, as previously mentioned, the manuscript requires polishing. Currently, it contains too many undefined terms, unclear parts, and disjointed sections, with little visibility on the main objectives and results. This makes it difficult to follow and could be problematic for Nature Communications, which aims at a non-expert audience. Therefore, I strongly recommend that the authors polish the manuscript before further consideration. I have provided some suggestions below on how it can be improved.

A lot of the critical terms are not defined: N, P, fixed nitrogen ... The same is true with key concepts (e.g., N:P ratio, P excess, the effect of upwelling on $\delta^{15}\text{N}$, increase burial decrease $\delta^{15}\text{N}$, ...). While these aspects need more descriptions, the manuscript still lacks conciseness, with a lot of concepts introduced but clearly connected to the aim of the paper and the main results. I suggest that the authors to identify more what the key results are and to focus on the elements that directly support these results, making sure that they are well-connected. Also, it is important to justify the aim (why focusing on the last 165 Myrs) and approach (why using models and proxy together?).

I also suggest that you make a diagram of the different $\delta^{15}\text{N}$ records ($\delta^{15}\text{N}_{\text{ngyre}}$, $\delta^{15}\text{N}_{\text{margin}}$, $\delta^{15}\text{N}_{\text{no3}}$...) and the processes that affect them.

There is a lot of references to geological periods, but without giving the dates, which is difficult to follow as most figures refer to dates and not time periods. Can you indicate dates each time you give a new time period?

Similarly, you discuss the evolution in paleogeography, but do not show it. It would help to be able to see map with the different ocean basins through the geological time. I suggest that you show the climate model upwelling outputs at the same time (at least in the SM).

One of the most interesting results is the effect of burial on $\delta^{15}\text{N}$ when upwelling increases. At the moment, you present it as a close-open system difference. It would be interesting to determine the level of OM burial required to change the trend in $\delta^{15}\text{N}$ (going from decrease to increase with increasing upwelling).

Specific comments:

The abstract needs polishing, as disjointed and unclear. Also it needs to clearly state why this study is important (reporting N and C isotopes over the last 165 Myrs), and what the method is. It is also hard to tease apart what you find and what you speculate. Please make this more clear. Also, there seems to be a series of key results, that are not well connected.

Replace "remote areas" and "NGyre" by "open ocean"

Need to justify more the assumptions to represent open ocean versus coastal regions in the box model (close versus open system for P), Also what about river inputs?

P24: Is 3.2 the modern ocean value? Please specify

P26-27: Not clear

P27-28: Is that one of your findings or a speculation. Which evidence do you provide for this?

P29-28: Not clear

P30: Say if the model has N cycling

P31-32: If the paleogeography is key, you need to introduce this concept more strongly and describe how it has changed over the last 165 Mys.

P37-39: This seems key, but not well introduced. Can you provide more elements to get to this conclusion?

P39-40: No real discussion about iron in the abstract. Can you expand on this?

P72-74: Not clear for non-experts - explain the concepts of global versus regional sources

P78: Explain why N:P ratio is an important metrics for your study

P78-79: That seems important. Can you explain it more? I am not sure I fully understand the relationship between change in climate, water formation, wind and the N:P ratio.

P93-94: Can you give the bigger implication of this aim, relating it to the iron cycle, and the ultimate nutrient?

P109-111: Can you comment on what the geological records might be under scenario of high or low N_2 fixation/denitrification?

P114-118: Not clear, too long sentence. Can you explain more? This seems crucial to your approach.

P119: $\Delta^{15}\text{N}$ not define

P121: Replace from by "measured by Kast et al."

P121-124: Not clear why NFB Ngyre and NNO_3 are different. Can you provide a schematic of the different processes, areas and isotropic signature to help?

P127-128: Awkward

P135-136: Not clear why you make this assumption as without burial you probably assume remineralisation of P.

P145: Can you show the data and define SD?

Fig.1: Cannot see NMarg, Ngyre clearly - is carbonate sites Ngyre? What about foram? Also, "very quick discussion" in the figure caption does not make sense. Besides, need to say more about $\delta^{15}\text{N}$. Why AIR? Which data did you measure and what is the difference between marginal and carbonate site? Finally, can you show which one are the new data (at least in the supplemental material)?

P148-150: Not clear. Please explain more

P154-155: Except for the drop at about 140 Myrs. Can you comment on this?

Fig

P158-159: I cannot see that. The values at 42 look more or less 3, so the same and not lower

P162: Not clear

P162-163: Need to introduce this concept better

P167-168: Show data

Fig.2: Give the equation of $\delta^{15}\text{N}$ (at least in the SM). Also, the scale for the changes in sea level is too large, hard to see

P172: not clear

P197: Need to justify why focusing on coastal upwelling (and ignore equatorial upwelling or deep water mixing)

P228-229: The ocean interior circulation will also influence the O_2 level and distribution in the ocean, which will affect dysoxic region and denitrification.

P235-236: Not clear

P237-238: Would be good to show the climate model surface upwelling to show the values regionally (in the SM)

P239: Replace conserved by P cycle is in a closed system. Replace left-hand panels with A and C ...

P241-242: Is that always the case or an assumption in your model?

P243: Say close and open P systems

Fig.3: Why A, B show upwelling and C/D show Transport. Can you be consistent or explain the difference? Also, say how you estimate global coastal upwelling flux (sum or average).

P254: Presumably in an open system, this would not be the case if river input from weathering is higher than upwelling. Can you comment on this?

P257: Can you show that as this is not intuitive? Also, is it because of enhanced burial (and not just increase EP)?

P277-278: Why do you get N limitation if there is N₂ fixation to compensate the N supply?

P290: Can you explain this concept? I thought primary producers would take the lighter isotopes so burial would tend to remove it from the system, resulting in an accumulation of heavy N in the water column.

P292-293: another coupling that you don't mention is that remineralisation should depend on oxygen level, reduced with lower oxygen concentration. There should then be a feedback where increasing upwelling causes less remineralisation than more burial.

P305-308: you keep referring to it, can you show maps of the evolution of continents/ocean configuration over time

P324-325: say why

P304: not clear what the main point of this section is. Can you expand?

P330-340: Not clearly explained the interplay between sedimentary denitrification, sea level rise and nitrogen fixation and 15N

P342-343: not clear

P344-345: Need to link this part better with your previous results

P358-368: This paragraph does not fit well here. You could move it with the early section detailing 13C record.

P386-395: Link this comment better with your key results

P399: what about the OM burial? This seems as important

P401: say when + show on the map

P477: limited by iron?

Version 1:

Reviewer comments:

Reviewer #1

(Remarks to the Author)

The authors have addressed many of my previous comments. I am still intrigued by the overall linkages that the paper uncovers. However, a few concerns or questions remain that need to be addressed:

In my previous review, I asked whether some of the selected sampling sites could have been restricted basins during the time of deposition. The authors replied that this was not the case, which is good. However, the response was not backed up by any references. I would like to see a new section of text in the supplementary material discussing the paleogeographic setting of the cores with a brief explanation of what is known about their connectivity to the open ocean. Adding such a section would significantly elevate the conclusions from this study and pre-empt potential doubt among future readers.

I am also still confused by the discussion of oxygen-minimum zones (OMZ). The box model includes OMZs; however, those are seemingly not part of the interpretation of the sedimentary data. The authors' response to my previous comment about d¹⁵N values in OMZs was that their sediment sites are not affected by OMZ effects. This creates a disconnect with the box model, where OMZs feature prominently. I would like to see more information about the sampling sites supporting the notion that OMZ processes (which create d¹⁵N values that are not globally representative, Tesdal et al. 2013) are not relevant to the sampling sites. This could also be added to the supplementary material. And/or perhaps the model needs to be adjusted to capture processes that are more relevant to the sampling sites instead.

Note that this may also be a misunderstanding, which may be resolvable with a better integration of the box model into the manuscript (see detailed comment below).

Line comments:

Throughout: I suggest changing N-fixation to N₂-fixation, by analogy to CO₂-fixation.

l. 57: delete the comma before 'nitrogenase'

l. 83: change inorganics to inorganic

l. 99: change 'on10 Myr' to 'in 10 Myr'

I. 106: nutrient-rich

II. 117: "Sediments underlying anoxic water are more efficient at retaining N" – a reference is needed at the end of this sentence.

II. 136-137: "The organisms that tend to dominate N-fixation fluxes occur in regions with warm sea surface temperatures" – a reference is needed here.

I. 158: change 'land based' to 'land-based'. However, 'land-derived' might fit better.

I. 161: "indicated by a smaller deficit relative to P (Fennel, 2010)." – provide the N/P ratio of upwelling regions or outer shelf regions as a quantitative reference point. Also correct the format of the reference.

II. 161-163: As noted in the previous round of reviews, areas with active upwelling often have higher d15N values than the global average (Tesdal et al. 2013 Biogeosciences). So this statement here needs to be qualified with additional information.

I. 170: change to 'foraminifera-bound'

II. 161-170: Please provide modern average values for d15N_DISTAL, d15N_DEEP, d15N_FB and D15N, so that the reader has a modern reference point in their head before reading about the results from the paleorecord. The modern data would provide important context for understanding these concepts.

Figure 2: I recommend plotting this figure in color. In the first panel with the d15N data, indicate which samples are deep and which ones are distal.

II. 223-224: 'indicates that vertical mixing increased'

I. 229: check the last sentence in this line. The structure is grammatically wrong. I think, the word 'converge' might need to be deleted.

II. 245-246: is there a unit of area (cm² or m²) missing in the carbon burial units?

I. 252: "Data for sea level and P accumulation rates (Fig. 2)" – this sentence is incomplete. What about those data? Please present them.

Fig. 3: What does PI in the figure legend mean? Please provide more information in the figure caption about what each of these models represent.

Fig. 3: It seems that the lines in the plot don't actually end at 0 Ma but instead around 5 Ma. What are the modern values that correspond to 0 Ma? Can those be indicated in the plot?

II. 290-366: This section is not well connected to the previous part of the paper. Figure 4 talks about OMZ versus DEEP sites, although the preceding discussion talked about DISTAL versus DEEP. The d15N behavior of OMZs (which are enriched in d15N compared to the deep ocean, see Tesdal et al. 2013), was not previously mentioned. Throughout this section here (II. 290-366) it is therefore unclear how the box model is going to be linked to the paleorecord. Please provide a better introduction to this section and state which open questions about the paleodata are going to be addressed with this box model. Explain to the reader how the OMZ and DEEP sites modelled here compare to the DISTAL and DEEP sites inferred from the sedimentary record.

I. 371: fall instead of falls. (the word 'data' is plural)

I. 382: "there are regions where excess P occurs in surface water, suggesting otherwise" – please provide a reference.

I. 438: indicate

I. 464: d15N_MARG appears here for the first time. Which record is used to reconstruct this, and what does it mean? How does it compare to DISTAL, DEEP and OMZ, which were used earlier?

I. 512: Please add a conclusion section that summarizes the key points from this study. The models and the paleodata each reveal new insights and relationships, and it would be nice to see those summarized here, perhaps along with an outlook to broader implications.

Eva Stüeken

(Remarks on code availability)

I'm not able to open the file, because I'm not familiar with FORTRAN. But I appreciate that the code is now available and will be useable by others in the future.

Reviewer #2

(Remarks to the Author)

I have reviewed the revised manuscript, and find it has been significantly improved, addressing most of my previous comments. I now recommend it for publication.

In particular, the addition of Figures 1 and 6 is effective in illustrating the evolution of paleogeography and conceptualising your systems. The discussion is also clearer and better structured, highlighting key results and providing key context, especially towards the role of iron limitation.

I have a few final comments on the reviewed version:

- Please define N and P in the abstract and intro.
- L252: The sentence is incomplete; please revise.
- L379: Typo - "increase" should be corrected to "increase".
- L444: The meaning is unclear; please clarify.

Lastly, I apologise for the delay in my review. The response to my comments lacked sufficient detail, in particular in specifying the lines where revisions were made. For future reference, I strongly recommend providing a more detailed response, as this would greatly help reviewers with busy schedule like mine :)

(Remarks on code availability)

Reviewer #3

(Remarks to the Author)

Review for Godfrey et al. Stability of the marine nitrogen cycle over the past 165 million years. Godfrey presented several new and published d15N (FB-d15N) and d13Corg records for the 165 million years. Using marginal d15N record to infer deep nitrate d15N changes, and open ocean d15N record as upper ocean nitrate d15N, they argue that the difference between the two is driven by vertical mixing, which is in turn, as they model exercise show, controlled by continental configuration. I appreciate their effort to generate this long dataset and the conduct of the modeling exercise to put these data into understanding. However, I disagree with a few key concepts used in this paper, and I also find that this paper is structurally challenging to follow.

1. The mean ocean nitrate d15N (or the deep ocean nitrate d15N) is determined by the ratio of N losses occurring in the sediments vs. in the water column, not by the processes occurring in the water column alone. Continental configuration should not only change upwelling, but perhaps more, to the N losses occurring in shelf sediments. In many places in this paper prior to discussion, the authors emphasized on the influence of water column denitrification on nitrate d15N, which is misleading.
2. I do not agree that d15N measured at the outer shelf equals deep nitrate. In response to the first reviewer, the authors explained that river influence is minimized by processes occurring in the inner shelf, and denitrification is not uniformly occurring. But even if these are the case, the shelf upwelling hardly works like a tunnel where deep nitrate is being directly upwelled and completely assimilated. If we look at modern ocean, places where strong upwelling occurs typically have unutilized nutrient left at the surface, so the degree of biological consumption drives most of the d15N changes over time. In other more oligotrophic places, marginal upwelling is still heavily influenced by upper ocean processes, such as nitrification, N2 fixation, denitrification. And I also think the d15N values of these records speak to a different story. The average d15Ndeep (values measure at the margins) is 3.2, similar to and lower than modern deep nitrate d15N (5.5). If the marginal d15N does reflect lower deep ocean nitrate d15N, then it calls for less water column denitrification comparing to benthic denitrification. But the distal d15N are much higher, averaged around 7 permil, which in the present ocean, can be only explained by water column denitrification.
3. Statistical treatment of these records. How did the authors treat these multiple records with different proxies (foram bound or bulk?), locations, temporal resolutions? The authors state that FB-d15N is higher than bulk because FB-d15N neglects N2 fixation (I disagree, see my comments below), but when calculating the differences among these records, did the authors simply take the average or pick one proxy over the other?

Line 77. Clarify the processes that cause N to be lost preferentially over P in the marginal sea. Do the authors mean redfield uptake of N and P and burial in sediments, or denitrification in shallow sediments?

Line 87-89. The average d15N of ocean nitrate is determined by the ratio of N losses through water column or sedimentary denitrification. It is not determined by the upwelled nitrate along the margins. I do not know what the authors mean here. The marginal upwelling might affect the mean d15N value if upwelling drives sedimentary denitrification, and assuming that water column denitrification remains constant through time. But the statement here made by the authors is wrong.

Line 161-163. Nitrate d15N in the outer shelf should not equal deep ocean even if river influence has been removed in the mid and inner shelf.

Line 172-174. Foram-bound d15N may also be influenced by newly fixed N, when the fixed N goes into the food web.

Overall FB-d15N shows strong correlation with thermocline nitrate, because insitu N2 fixation is a small flux comparing to vertical supplies. It is the same as sinking PON. The d15N of sinking PON, and thus buried bulk organic N in the sediments, should reflect the d15N of all the new sources of N to the euphotic zone. But comparing d15N of bulk sinking PN with these new sources, there is a stronger influence from vertical supplies, and not from insitu N2 fixation. So I would not say foram-bound d15N and bulk d15N differ in this.

Line 255, "where d15N of the ocean is defined". I guess the authors mean the deep ocean nitrate d15N. However, the deep

ocean nitrate d15N is defined by the ratio between N loss occurring in the sediments vs. in the water column, not by the processes in the water column alone. This is a confusing sentence.

Figure 1. I think the locations of 1263 and 1209 are wrong.

Line 226. D15NFB captures the d15N signature of nitrate below the mixed layer, but tends to neglect contributions from N fixation. See my previous comments. I do not think this is the case, at least there is no evidence for this statement.

The authors use d15N_{distal} and d15N_{gyre} to refer to the same thing, and use d15N_{margin} and d15N_{deep} to refer to the same thing, but need a statement at least when introducing the terms.

Line 353. The lighter the oceanic N inventory becomes? I guess the authors mean the lighter the oceanic d15N becomes.

(Remarks on code availability)

Version 2:

Reviewer comments:

Reviewer #1

(Remarks to the Author)

The authors have largely addressed my previous comments. I just noticed a few points that should be clarified before publication:

I. 63: It is not necessary to keep the name of the first author when using numbered references. This looks odd.

I. 68: N₂-fixers

II. 73-76: references needed

II. 91-96: I worry that this section is difficult to understand for readers who are not familiar with these concepts. It would help if the isotope mass balance were presented in more quantitative terms. State what percentage of fractionation is taking place in the water column and what percentage takes place in sediments. Also add here that sedimentary denitrification imparts limited net isotopic fractionation. Otherwise, it might be unclear to readers why the balance between water column and sedimentary denitrification is important.

I. 126: fixed N inventory (to exclude dissolved N₂)

I. 128: delete 'either'

II. 138-140: The importance of euxinic pore waters should be emphasized here. Under ferruginous conditions, phosphate can be trapped in ferrous phosphate minerals (vivianite and Fe-rich clays). However, the presence of free sulfide traps ferrous iron in pyrite, enabling phosphate to escape more efficiently into the water column (see review by Guilbaud 2025 *Treatise in Geochemistry*)

II. 149-151: This sentence seems out of place and is probably not needed here. I suggest deleting it.

I. 157: change to 'hypoxic or anoxic' so that it cannot be mistaken as a ratio.

I. 166: delete the comma after 'reservoir'. As noted above, consider moving this information about fractionations upwards.

I. 172: is this m³ of water or of sediment?

I. 182: nutricline

I. 197: Add a sentence here to clarify if d15N_{DISTAL} was also measured somehow, or modelled, or if this term is just a theoretical concept. This is unclear.

I. 212: What is 'it'? The composition of DIC? Please rephrase.

I. 240: Referring to my earlier comment about I. 197, please explain how d15N_{DISTAL} was constrained from the data.

Figure 2: This figure is difficult to read. Increase the font size and use a sans serif font, such as Arial or Calibri. The symbols are also difficult to match up with the legend, because they are quite small. In the figure caption, black and green symbols in the d15N curve are explained, but not the blue and orange symbols. What are those?

I. 282: Define d15N_{GYRE} and explain how it was determined. This is unclear.

I. 304: replace 'dramatically' by a more scientific, quantitative term, such as 'by a factor of XX'.

l. 428: N2-fixers

l. 435: suggests

Figure 7: As for Figure 2, the font is difficult to read. Change to a sans serif font such as Arial or Calibri. I suggest making it consistent with the other figures.

l. 549: N2-fixers

l. 569 and 572: superscript in d15N

Eva Stüeken

(Remarks on code availability)

I'm not familiar with Fortran and have therefore only read through the description of the model in the supplements. This looks reasonable and is explained well.

Response to reviewer comments

Reviewer #1 (Remarks to the Author):

The authors present new nitrogen and carbon isotope data from marine sediments spanning from 165Ma to the modern and use these data, along with a computational box model, to assess linkages between coastal upwelling, biological productivity, and P and N cycling. The results reveal previously undocumented feedbacks between physical oceanographic conditions and biogeochemical cycling that help explain patterns in the data that were previously not well understood.

I have spent a lot of my time thinking about linkages between nitrogen isotopes, redox and productivity, and I am excited to see how these parameters can be linked to global oceanographic and orogenic processes. Many patterns in paleoenvironmental proxy data have always been difficult to explain by biogeochemistry alone, and so this linkage to physical and geological phenomena that operate on a global scale is refreshing. I could imagine that these findings will lead to new hypothesis about biogeochemical cycles even further back in time.

I do, however, have some comments that should be taken into account to strengthen the conclusions.

1) Despite the extensive effort to accumulate such a large dataset, I worry that the data are not globally representative, given limitations of availability. According to Fig. 1, several of the sites appear to be located in possibly restricted basins. In the modern ocean, restricted basins can be distinct in $\delta^{15}\text{N}$ from open marine margins. The supplementary file needs a more detailed description of the major sampling sites to resolve this concern.

We acknowledge this is a concern studying marine sediments when much of the oceanic crust has been subducted, but the paleogeography indicates that the sites were not located in small shallow basins, but large ones, even as Tethys opened.

2) I'm not entirely convinced by the assumption that $\delta^{15}\text{N}$ along coastal margins captures deep marine nitrate. Tesdal et al. 2013 (Biogeosciences) show that the highest $\delta^{15}\text{N}$ values in sediments tend to coincide with upwelling zones. Here, denitrification rates are locally high, causing dissolved nitrate to be locally enriched in $\delta^{15}\text{N}$ (up to +12permil, in fact consistent with some of the model results in Fig. 3), but this is likely not representative of the global deep marine nitrate pool. It would help if modern data could be used to illustrate the overall concept of comparing marginal to gyre sediments as a proxy for nitrate advection.

The large OMZs of the ocean do occur at margins due to upwelling, but not every margin is an upwelling zone. The reviewer highlights that a better way of describing the nutrient processes of the pre-industrial outer shelf is nutrient onlapping from the open ocean (see added reference Fennel,

2010). Shelf sediments act as filters for nutrients associated with run-off, even in the industrial era, so that balancing the N budget today requires large overlapping fluxes of N. Due to the proximity of the coast, the availability of P and micronutrients mean that N is the limiting nutrient and it will record the isotope composition on the fixed N. Sediment denitrification does not have associated isotope effect on the water column, and relatively rapid burial makes these sediments less prone to diagenetic effects of oxidation near the sediment water interface. Tesdal et al., 2013 also provide a map of surface sediments showing that away from the anoxic water columns of the major upwelling zones, margin sediments are between 4 and 6 per mil, the value of the deep ocean fixed N. I also provide data in the Appendix for a transect of core top material at Cape Hatteras which shows that while organic C bears some trace of land-derived organic C, the N isotope data are invariant from 400 to 3000m water depth. This is all more described in the text.

3) It makes a significant difference in the model results whether P is treated as static or dynamic, but it is not made clear how these two scenarios can be gleaned from the paleo-record. When discussing the model results later on, more information (ideally quantitative) is needed for how the state of P can be constrained.

Although there are time periods and ocean basins within which P is more conserved than in others, there will always be some input and output of P. In other words, a dynamic P cycle is simply more realistic than a static ocean P inventory, regardless of the paleo-record. The reason why we included simulations with the static P model is to illuminate causes and effects. In particular, the static P model includes only one key mechanism for changing ocean $\delta^{15}\text{N}$: water-column denitrification. Thus, the static-P simulations allowed us to focus on this mechanism in isolation. The dynamic P model includes both water-column denitrification and burial as mechanisms for changing ocean $\delta^{15}\text{N}$. Hence, the reversed trend in the dynamic P model has to be caused by burial. We have included this explanation in the revised manuscript.

Minor points:

ll. 74-75: “We explore whether productivity followed P input to the surface ocean” – please explain on which timescales you are exploring this and by which means. This sentence actually seems to be out of place. Maybe delete it or move it further down where you transition to describing the aims of this study.

We agree, so one sentence is deleted and the other modified.

l. 78: Measured N:P ratios in what? I assume in seawater, but it could also be in biomass. Please clarify in this instance.

This has been clarified

L. 121: Are $\delta^{15}\text{N}_{\text{GYRE}}$ and $\delta^{15}\text{N}_{\text{NFB}}$ are similar when the foraminifera are obtained from gyre regions as well? Or do foraminifera from any part of the ocean match $\delta^{15}\text{N}_{\text{GYRE}}$? Please clarify.

Yes, it is related to sampling locations, this has been clarified in the text.

l. 123: “in the upper thermocline” – this presumably depends on the species of foraminifera and their preferred habitat, or not?

True, but most foraminifera are located where there is food, so the ones studied feed at this depth.

Fig 1 and associated text (ll. 144-153): The text states that both marginal and gyre sites are shown in Fig. 1; however, the figure’s legend lists margin sites, carbonate sites and foraminifera bound samples. Gyre samples are not mentioned in the legend. Please highlight those. Carbonate samples have not been mentioned in the text until now. It would be good to explain how those compare to the margin and gyre sites. Amendment: With further reading, I get the impression that the carbonate sites are the gyre sites. That confuses me, because a lot of carbonate precipitates in shallow waters. So some clarification is needed.

The figure has been corrected. We renamed the carbonate sites gyre, but this figure legend did not get updated.

Fig. 1 C-accumulation: Change the axis labels into a more readable notation instead of Excel’s version (e.g. 10^{-4} instead of 1.E+04). Also, should the units not have a surface area in them? (e.g., g/kyr/m² or something like that?). If this is a global C burial rate, then the values seem very small.

This figure has been replotted.

ll. 236-237: something is wrong with this sentence. Please rewrite.

We have removed this confusing sentence, which did not add much useful information.

l. 361: demand for CO₂ instead of demand for C?

yes

ll. 304-399: This review section seems a bit out of place. It made sense to have the modelling after the presentation of the data, but after reading about the model, I expected to see the model results linked to the data presented above. But that’s not the case. Instead, the manuscript continues with a more general review of other topics, and the linkage between the data and the model does not occur until l. 400 onwards. I found that confusing. Please consider reorganising the text. The review portion between ll. 304-399 could probably be shortened and perhaps move up or merged with later parts of the discussion.

WORKING ON IT

L. 423: How does one know if sediments play an important role in nutrient storage? This requires explanation.

We address this by the org-C, P and to a lesser extent, N accumulation rates. While a sink for N maybe the atmosphere, the only sink for P is burial. The importance of burial is considered through the use of 2 models.

ll. 448-450: This sentence seems to contradict the previous one, according to which upwelling over deep water caused efficient remineralization of OM before settling on the seafloor (and thus presumably low OM accumulation). If that was the case, then the P inventory must not necessarily have been lower to explain low OM accumulation.

Remineralization depths for P and N are different you're right, globally more efficient recycling – but we talking about local effect so we cannot address inventory. We removed the sentence.

l. 469: It would be good if quantitative temperature thresholds could be provided for N-fixers and for the inferred water conditions.

Methods: State the reproducibility of samples and quality control standards

This is in the appendix

Which software was used to set up the model? It would be ideal if the code could be shared in the appendix.

The model was coded in FORTRAN. We have now included the source code including instructions as online Supplementary Material. Furthermore, the source code is available through Omta's GitHub page.

Please share an Excel spreadsheet with all the new sedimentary N and C data.

The file should have been transferred with the manuscript, it is submitted here.

Eva Stüeken

Reviewer #2 (Remarks to the Author):

This study aims to analyse the relationship between the nitrogen and carbon cycles over the last 165 Myrs. It presents new nitrogen and carbon isotope data from a series of cores. It combines it with a global climate model to reconstruct ocean upwelling and a box model to characterise the dynamics between the nitrogen isotopes, upwelling and production. This study presents some interesting and potentially important results, including the role of organic matter burial, upwelling and iron on the nitrogen isotope values and the implications for long-term nitrogen and carbon cycles. However, the paper covers many aspects that need to be better connected, making it challenging to identify the key findings.

The paper has the potential to have significant results for Nature Communications. It presents the first long-term record of nitrogen and carbon isotopes and combines modelling to examine the controlling processes. However, as previously mentioned, the manuscript requires polishing. Currently, it contains too many undefined terms, unclear parts, and disjointed sections, with little visibility on the main objectives and results. This makes it difficult to follow and could be problematic for Nature Communications, which aims at a non-expert audience. Therefore, I strongly recommend that the authors polish the manuscript before further consideration. I have provided some suggestions below on how it can be improved.

A lot of the critical terms are not defined: N, P, fixed nitrogen ... The same is true with key concepts (e.g., N:P ratio, P excess, the effect of upwelling on $\delta^{15}\text{N}$, increase burial decrease $\delta^{15}\text{N}$, ...). While these aspects need more descriptions, the manuscript still lacks conciseness, with a lot of concepts introduced but clearly connected to the aim of the paper and the main results. I suggest that the authors identify more what the key results are and to focus on the elements that directly support these results, making sure that they are well-connected. Also, it is important to justify the aim (why focusing on the last 165 Myrs) and approach (why using models and proxy together?).

I also suggest that you make a diagram of the different $\delta^{15}\text{N}$ records ($\delta^{15}\text{N}_{\text{gyre}}$, $\delta^{15}\text{N}_{\text{margin}}$, $\delta^{15}\text{N}_{\text{no}_3}$...) and the processes that affect them.

There is a lot of references to geological periods, but without giving the dates, which is difficult to follow as most figures refer to dates and not time periods. Can you indicate dates each time you give a new time period?

We use dates rather than periods in the text now, but the period is included with date in figure 2.

Similarly, you discuss the evolution in paleogeography, but do not show it. It would help to be able to see map with the different ocean basins through the geological time. I suggest that you show the climate model upwelling outputs at the same time (at least in the SM).

We created a new Figure 1, which is more clear than the maps that had been part of fig 1 in the first version.

One of the most interesting results is the effect of burial on $\delta^{15}\text{N}$ when upwelling increases. At the moment, you present it as a close-open system difference. It would be interesting to determine the level of OM burial required to change the trend in $\delta^{15}\text{N}$ (going from decrease to increase with increasing upwelling).

Yes indeed. I think for such an exercise to produce precise numbers, more cores would need to be included due to the heterogeneity of accumulation. However, rather than strictly accumulation rates that switch between open and closed behaviour, it is the geometry of the ocean basins, where upwelling occurs in relation to where new N can be added (N-fixation). This is discussed.

Reviewer 2 makes important comments that we address as follows.

As with Reviewer 1's comments the discussion is kept on point by unifying the model with the data plus paleoenvironment. We give a more thorough description of the terms based on a non-specialist reader, and lay them out earlier in the paper.

165Myr was chosen based on the availability of continuous core, and that this study follows from earlier ones (Katz et al., Falkowski et al) using virtually the same material from DSDP/ ODP/IODP collections.

Dates and time periods are provided.

The world ocean basin evolution is shown in Fig 1., the evolution of the different basins with core locations are provided in the supplement.

Specific comments:

The abstract needs polishing, as disjointed and unclear. Also it needs to clear state why this study is important (reporting N and C isotopes over the last 165 Myrs), and what the method is. It is also hard to tease apart what you find and what you speculate. Please make this more clear. Also, there seems to have a series of key results, that are not well connected.

Replace "remote areas" and "NGyre" by "open ocean"

Done

Need to justify more the assumptions to represent open ocean versus coastal regions in the box model (close versus open system for P), Also what about river inputs?

The open model just adds P and is agnostic as to where it originates from.

P24: Is 3.2 the modern ocean value? Please specify
It's the 165 My average, text has been changed

P26-27: Not clear

A sentence was added earlier, and this sentence has been clarified.

P27-28: Is that one of your findings or a speculation. Which evidence do you provide for this?

We add a qualifier, but discuss more fully with evidence in the text.

P29-28: Not clear

Addressed

P30: Say if the model has N cycling

There are 2 models, the one that determines the relative strength of upwelling is a physical model.

P31-32: If the paleogeography is key, you need to introduce this concept more strongly and describe how it has changed over the last 165 Mys.

This is addressed in the text, not the abstract.

P37-39: This seems key, but not well introduced. Can you provide more elements to get to this conclusion?

Done

P39-40: No real discussion about iron in the abstract. Can you expand on this?

Done

P72-74: Not clear for non-experts - explain the concepts of global versus regional sources

This is expanded.

P78: Explain why N:P ratio is an important metrics for your study

Done

P78-79: That seems important. Can you explain it more? I am not sure I fully understand the relationship between change in climate, water formation, wind and the N:P ratio.

The importance is included here, but the relationships are considered in depth shortly after this part.

P93-94: Can you give the bigger implication of this aim, relating it to the iron cycle, and the ultimate nutrient?

Done

P109-111: Can you comment on what the geological records might be under scenario of high or low N₂ fixation/denitrification?

By addressing the role of productivity and how N₁₅ will respond if it is open or closed nutrient system, we can determine when nutrients were conserved (closed) or buried (open). For most of the last 165 Myr, the latter provides a better model of the system.

P114-118: Not clear, too long sentence. Can you explain more? This seems crucial to your approach.

This sentence has been rewritten.

P119: Delta15N not define

This is now defined

P121: Replace from by “measured by Kast et al.”

Replaced

P121-124: Not clear why NFB Ngyre and NNO3 are different. Can you provide a schematic of the different processes, areas and isotropic signature to help?

A new figure has been created.

P127-128: Awkward

Rewritten

P135-136: Not clear why you make this assumption as without burial you probably assume remineralisation of P.

Yes, and this is made explicit in the text.

P145: Can you show the data and define SD?

SD is standard deviation. The data is in an accompanying excel file.

Fig.1: Cannot see NMarg, Ngyre clearly - is carbonate sites Ngyre? What about foram? Also, “very quick discussion” in the figure caption does not make sense. Beside, need to say more about d15N. Why AIR? Which data did you measure and what is the difference between marginal and carbonate site? Finally, can you show which one are the new data (at least in the supplemental material)?

Nitrogen isotopes are reported relative to the atmosphere which is commonly referred to as “AIR”. The marg and gyre terms were renamed and are consistent now with the text. The caption has been rewritten, it seems to have notes to myself on it.

P148-150: Not clear. Please explain more

This is clarified in the text

P154-155: Except for the drop at about 140 Myrs. Can you comment on this?

There are two times when N15 was low, around 162 and again 140 Myr. These are discussed in the discussion section rather than the presentation of the data and model, but these drops are now

noted here.

Fig

P158-159: I cannot see that. The values at 42 look more or less 3, so the same and not lower

I agree, this is deleted.

P162: Not clear

Rewritten.

P162-163: Need to introduce this concept better

We rewrote the description of this concept P140-158

P167-168: Show data

It is included on Figure 1.

Fig.2: Give the equation of $\delta^{15}\text{N}$ (at least in the SM). Also, the scale for the changes in sea level is too large, hard to see

We have endeavoured to expand the sea level scale. The equation for delta notation is now written in the text P116-117

P172: not clear

Expanded.

P197: Need to justify why focusing on coastal upwelling (and ignore equatorial upwelling or deep water mixing)

The denitrification that affects the N isotope composition of the ocean occurs along areas of boundary upwelling which have greater water flux than equatorial upwelling. Consequently we focus on these areas. However, the isotope model is not dependent on where denitrification occurs.

P228-229: The ocean interior circulation will also influence the O_2 level and distribution in the ocean, which will affect dysoxic region and denitrification.

That is true, we have corrected this.

P235-236: Not clear

We have removed this confusing sentence, which did not add much useful information.

P237-238: Would be good to show the climate model surface upwelling to show the values regionally (in the SM)

P239: Replace conserved by P cycle is in a closed system. Replace left-hand panels with A and C ...

We have incorporated this.

P241-242: Is that always the case or an assumption in your model?

This is a model assumption that we made to approximate the real P cycle. That is, weathering is the main source and burial is the main sink of oceanic P (Filippelli, 2002). We have included this motivation in the revised manuscript.

P243: Say close and open P systems

We have implemented this.

Fig.3: Why A, B show upwelling and C/D show Transport. Can you be consistent or explain the difference? Also, say how you estimate global coastal upwelling flux (sum or average).

The horizontal axes in panels A, B, C, and D all indicate the same quantity, our apologies for the confusion caused by the different labels. We have made the labels consistent. The coastal upwelling mass flux in the upper panel of Fig. 3 represents the globally spatially integrated upwelling transport, i.e., the sum. We have specified this in the caption of Fig. 3.

P254: Presumably in an open system, this would not be the case if river input from weathering is higher than upwelling. Can you comment on this?

Even if upwelling accounts for only a minor fraction of the P input to the ocean surface, increasing upwelling will lead to increased export production (even if only marginally).

P257: Can you show that as this is not intuitive? Also, is it because of enhanced burial (and not just increase EP)?

Only burial can remove nutrients, EP does not.

P277-278: Why do you get N limitation if there is N₂ fixation to compensate the N supply?

P290: Can you explain this concept? I thought primary producers would take the lighter isotopes so burial would tend to remove it from the system, resulting in an accumulation of heavy N in the water column.

Although primary producers preferentially take up the lighter N isotope, there is essentially no fractionation due to export in the model because almost all the N in the surface box is consumed. As a result, exported N has approximately the same isotopic composition as the ocean average, which is isotopically heavier than N lost to water-column denitrification. Thus, the more N is lost due to burial or sediment denitrification, the lighter the oceanic N inventory becomes. In contrast, the more N is lost due to water-column denitrification, the heavier the oceanic N inventory becomes. We have included this explanation in the revised manuscript.

P292-293: another coupling that you don't mention is that remineralisation should depend on oxygen level, reduced with lower oxygen concentration. There should then be a feedback where increasing upwelling causes less remineralisation then more burial.

Enhanced upwelling increases export, which decreases the oxygen concentration in the OMZ. This will indeed enhance burial due to decreased remineralization. We have included a discussion of this effect in the revised manuscript.

P305-308: you keep referring to it, can you show maps of the evolution of continents/ocean configuration over time

These are shown in Figure 1, but maps specific to the core locations are now in the SM

P324-325: say why

P304: not clear what the main point of this section is. Can you expand?

This section had importance in describing the changing physical conditions, but we have shortened and moved up it into the initial description of the data.

P330-340: Not clearly explained the interplay between sedimentary denitrification, sea level rise and nitrogen fixation and ^{15}N

P342-343: not clear

P344-345: Need to link this part better with your previous results

P358-368: This paragraph does not fit well here. You could move it with the early section detailing ^{13}C record.

P386-395: Link this comment better with your key results

These comments are addressed by a significant rewrite and re-organization of the manuscript

P399: what about the OM burial? This seems as important

Yes, there is a new figure that investigates the role of OM burial (Fig 6)

P401: say when + show on the map

This addresses a part of the paper which has been deleted in an effort to focus the paper on mechanisms and not a history.

P477: limited by iron?

The model does not include Fe, but it assumes that Fe, or any other micronutrient, cannot inhibit N-fixation. The role of Fe limitation on N dynamics is covered (but not deeply) at the end of the paper, and clearly warrants further modeling, but is beyond the scope of this paper.

RESPONSE TO REVIEWER COMMENTS

Reviewer #1 (Remarks to the Author):

The authors have addressed many of my previous comments. I am still intrigued by the overall linkages that the paper uncovers. However, a few concerns or questions remain that need to be addressed:

In my previous review, I asked whether some of the selected sampling sites could have been restricted basins during the time of deposition. The authors replied that this was not the case, which is good. However, the response was not backed up by any references. I would like to see a new section of text in the supplementary material discussing the paleogeographic setting of the cores with a brief explanation of what is known about their connectivity to the open ocean. Adding such a section would significantly elevate the conclusions from this study and pre-empt potential doubt among future readers.

I am also still confused by the discussion of oxygen-minimum zones (OMZ). The box model includes OMZs; however, those are seemingly not part of the interpretation of the sedimentary data. The authors' response to my previous comment about $\delta^{15}\text{N}$ values in OMZs was that their sediment sites are not affected by OMZ effects. This creates a disconnect with the box model, where OMZs feature prominently. I would like to see more information about the sampling sites supporting the notion that OMZ processes (which create $\delta^{15}\text{N}$ values that are not globally representative, Tesdal et al. 2013) are not relevant to the sampling sites. This could also be added to the supplementary material. And/or perhaps the model needs to be adjusted to capture processes that are more relevant to the sampling sites instead.

Note that this may also be a misunderstanding, which may be resolvable with a better integration of the box model into the manuscript (see detailed comment below).

The description of the sample sites has been added to the supplement. We have tried to keep it succinct. The description of the model has had the role of the upwelling zones within the boxes expanded to improve clarity.

Line comments:

Throughout: I suggest changing N-fixation to N_2 -fixation, by analogy to CO_2 -fixation.

14 changes made

l. 57: delete the comma before 'nitrogenase'

deleted

I. 83: change inorganics to inorganic

changed

I. 99: change 'on10 Myr 'to 'in 10 Myr '

changed

I. 106: nutrient-rich

dash added

I. 117: "Sediments underlying anoxic water are more efficient at retaining N" – a reference is needed at the end of this sentence.

2 references added

II. 136-137: "The organisms that tend to dominate N-fixation fluxes occur in regions with warm sea surface temperatures" – a reference is needed here.

Reference added

I. 158: change 'land based 'to 'land-based'. However, 'land-derived 'might fit better.

Changed to "Continental shelves act as filters for bioavailable and organic N formed on land,...."

I. 161: "indicated by a smaller deficit relative to P (Fennel, 2010)." – provide the N/P ratio of upwelling regions or outer shelf regions as a quantitative reference point. Also correct the format of the reference.

Fennel uses N*, so we refer to that, and we do define it. Since she referenced the shelf values to the open Atlantic, that reference is also added

II. 161-163: As noted in the previous round of reviews, areas with active upwelling often have higher d15N values than the global average (Tesdal et al. 2013 Biogeosciences). So this statement here needs to be qualified with additional information.

This paragraph was expanded both in terms of qualifying the areas where high d15N is generated (the OMZs) and also the open ocean (which was driven by comments from Reviewer 3)

I. 170: change to 'foraminifera-bound '

The hyphen is added

II. 161-170: Please provide modern average values for d15N_DISTAL, d15N_DEEP, d15N_FB and D15N, so that the reader has a modern reference point in their head before reading about the results from the paleorecord. The modern data would provide important context for understanding these concepts.

This section has been expanded a little with typical values added and most of it has been moved into the end of the introduction rather than methods.

Figure 2: I recommend plotting this figure in color. In the first panel with the d15N data, indicate which samples are deep and which ones are distal.

The figure is in colour, and the caption highlights which data are distal vs deep

II. 223-224: 'indicates that vertical mixing increased '

The word "that" has been inserted

I. 229: check the last sentence in this line. The structure is grammatically wrong. I think, the word 'converge 'might need to be deleted.

"Converge" has been deleted

II. 245-246: is there a unit of area (cm² or m²) missing in the carbon burial units?

Correct. The /cm² has been added

I. 252: "Data for sea level and P accumulation rates (Fig. 2)" – this sentence is incomplete. What about those data? Please present them.

This has been fixed

Fig. 3: What does PI in the figure legend mean? Please provide more information in the figure caption about what each of these models represent.

PI is pre industrial. The legend has been clarified in the caption.

Fig. 3: It seems that the lines in the plot don't actually end at 0 Ma but instead around 5 Ma. What are the modern values that correspond to 0 Ma? Can those be indicated in the plot?

The figure has been amended, but it does end at 0, in the figure it has a label of PI, pre-industrial, the major ticks are on odd 10-Myr intervals

II. 290-366: This section is not well connected to the previous part of the paper. Figure 4 talks about OMZ versus DEEP sites, although the preceding discussion talked about DISTAL versus DEEP. The d15N behavior of OMZs (which are enriched in d15N compared to the deep ocean, see Tesdal et al. 2013), was not previously mentioned. Throughout this section here (II. 290-366) it is therefore unclear how the box model is going to be linked to the paleorecord. Please provide a better introduction to this section and state which open questions about the paleodata are going to be addressed with this box model. Explain to the reader how the OMZ and DEEP sites modelled here compare to the DISTAL and DEEP sites inferred from the sedimentary record.

Although the nitrate and N-15 signals are created within OMZs, they are responsible for the N-15 signature of the deep ocean as a result of mixing. One would not want to measure N-15 at an upwelling site to try and see its influence on the ocean-wide N cycle, because any one upwelling site could be un-representative on both spatial and temporal scales. Nevertheless, all OMZs combined will affect the deep water signature and this is the rationale for trying to get a record of deep-water nitrate. Text has been added to the start of the section to clarify this and the rationale of the 2 treatments of P has been expounded.

I. 371: fall instead of falls. (the word 'data' is plural)

This has been changed

I. 382: "there are regions where excess P occurs in surface water, suggesting otherwise" – please provide a reference.

two are provided

I. 438: indicate

fixed

I. 464: d15N_MARG appears here for the first time. Which record is used to reconstruct this, and what does it mean? How does it compare to DISTAL, DEEP and OMZ, which were used earlier?

d15N marg was changed to d15N deep, these old terms are changed

I. 512: Please add a conclusion section that summarizes the key points from this study. The models and the paleodata each reveal new insights and relationships, and it would be nice to see those summarized here, perhaps along with an outlook to broader implications.

We went back and forth about including an outlook based on the model and data, and decided that was probably beyond the scope of this paper. The final paragraph was increased in length to be a more typical conclusion/roundup.

Eva Stüeken

Reviewer #1 (Remarks on code availability):

I'm not able to open the file, because I'm not familiar with FORTRAN. But I appreciate that the code is now available and will be useable by others in the future.

Reviewer #2 (Remarks to the Author):

I have reviewed the revised manuscript, and find it has been significantly improved, addressing most of my previous comments. I now recommend it for publication.

In particular, the addition of Figures 1 and 6 is effective in illustrating the evolution of paleogeography and conceptualising your systems. The discussion is also clearer and better structured, highlighting key results and providing key context, especially towards the role of iron limitation.

I have a few final comments on the reviewed version:

- Please define N and P in the abstract and intro.

Done

- L252: The sentence is incomplete; please revise.

Revised

- L379: Typo - "increase" should be corrected to "increase".

Corrected

- L444: The meaning is unclear; please clarify.

This relates the 2 trends between d15N and Corg, and has been clarified in the text.

Lastly, I apologise for the delay in my review. The response to my comments lacked sufficient detail, in particular in specifying the lines where revisions were made. For future reference, I strongly recommend providing a more detailed response, as this would greatly help reviewers with busy schedule like mine :)

Sincere apologies to reviewer 2

Reviewer #3 (Remarks to the Author):

Review for Godfrey et al. Stability of the marine nitrogen cycle over the past 165 million years. Godfrey presented several new and published $\delta^{15}\text{N}$ (FB- $\delta^{15}\text{N}$) and $\delta^{13}\text{C}_{\text{org}}$ records for the 165 million years. Using marginal $\delta^{15}\text{N}$ record to infer deep nitrate $\delta^{15}\text{N}$ changes, and open ocean $\delta^{15}\text{N}$ record as upper ocean nitrate $\delta^{15}\text{N}$, they argue that the difference between the two is driven by vertical mixing, which is in turn, as they model exercise show, controlled by continental configuration. I appreciate their effort to generate this long dataset and the conduct of the modeling exercise to put these data into understanding. However, I disagree with a few key concepts used in this paper, and I also find that this paper is structually challenging to follow.

Text has been moved, and introductory sentences added. We agree that there are several different threads to the paper and the structure gets rather difficult. But having the model results early seemed best so that the discussion can draw on something already presented.

1. The mean ocean nitrate $\delta^{15}\text{N}$ (or the deep ocean nitrate $\delta^{15}\text{N}$) is determined by the ratio of N losses occurring in the sediments vs. in the water column, not by the processes occurring in the water column alone. Continental configuration should not only change upwelling, but perhaps more, to the N losses occurring in shelf sediments. In many places in this paper prior to discussion, the authors emphasized on the influence of water column denitrification on nitrate $\delta^{15}\text{N}$, which is misleading.

This is a first order view of the N cycle, but we agree that more could be said about the role of continent configuration and sediment denitrification. However, sea-level changes shelf width everywhere and clearly is not the long term driver of $\delta^{15}\text{N}$. The Scotese model indicates that the area of shallow seas is greatest when rifting starts, when foram-bound and distal $\delta^{15}\text{N}$ are highest. So it is clearly a complicated issue.

2. I do not agree that $\delta^{15}\text{N}$ measured at the outer shelf equals deep nitrate. In response to the first reviewer, the authors explained that river influence is minimized by processes occurring in the inner shelf, and denitrification is not uniformly occurring. But even if these are the case, the shelf upwelling hardly works like a tunnel where deep nitrate is being directly upwelled and completely assimilated. If we look at modern ocean, places where strong upwelling occurs typically have unutilized nutrient left at the surface, so the degree of biological consumption drives most of the $\delta^{15}\text{N}$ changes over time. In other more oligotrophic places, marginal upwelling is still heavily influenced by upper

ocean processes, such as nitrification, N₂ fixation, denitrification. And I also think the d¹⁵N values of these records speak to a different story. The average d¹⁵N_{deep} (values measure at the margins) is 3.2, similar to and lower than modern deep nitrate d¹⁵N (5.5). If the marginal d¹⁵N does reflect lower deep ocean nitrate d¹⁵N, then it calls for less water column denitrification comparing to benthic denitrification. But the distal d¹⁵N are much higher, averaged around 7 permil, which in the present ocean, can be only explained by water column denitrification.

Measurement of sediment d¹⁵N by shelves where there is not strong upwelling have values like the deep ocean. Measurements were done at different depths off Cape Hatteras show this, and Tesdale presented this too. There is no continuous or seasonal upwelling at the sites we have used. On these margins on-welling if water from currents like the Gulf Stream or offshore water masses supply the nutrients. The high productivity of these shelves partly comes from land, but there are older papers by Seitzinger (as well as the Fennel one we use) that shows land derived N is denitrified on the shelf. Tunnels or fire hoses are not needed, turbulent mixing will suffice.

Regarding the imprint of biological activity and N sources on shelves with Cape Hatteras as an example of one where there is not strong upwelling (the type we drew our record from):

- Fixed N is basically always drawn down completely in surface waters across the outer MAB/SAB shelf (many refs show this), so upper ocean recycling shouldn't have a significant impact on d¹⁵N as the reviewer suggests.
- Though N₂ fixation rates can be significant when compared on a volumetric basis to open ocean rates, N_{fix} generally supplies a very small fraction of the fixed N for primary production locally (until you get out into the slope sea or GS, then it matters a bit more) and, consequently, shouldn't have a major impact on bulk sediment d¹⁵N. Seldon et al., 2021 paper in L&O shows this across the Cape Hatteras frontal region.

Given these constraints, we do not know what 'story' would be told. The 3.2 pm average is long term, we do not think the N cycle has to have responded in a way to the different conditions over the last 165Myr to give nitrate in the deep ocean the same isotope composition as the modern ocean. Balancing the N-isotope budget of the modern ocean used best estimates of various terms, some of which were known to be rather imprecise due to the complexity of the N cycle, and from this the sediment denitrification versus N₂-fixation rates were estimated. So, it was not the situation that denitrification in either the water column or sediments was measured, it was estimated to resolve a balanced budget. The question arises as to what happens with the N budget if something like water column denitrification were to change – and how does the isotope composition of fixed N change as a reflection (assuming that the N budget is balanced). This is what

the paper aims to address. We do not assume that the upwelling systems have had the same influence on nitrate N15 through geologic time, and therefore one **only** has to change the ratio of denitrification rates in sediments versus water column.

The distal and FB N15 records speak more to stratification and the preservation and export of signals generated elsewhere. There is nitrate left in the surface ocean above OMZs, but the d15N signal is not as extreme as the values at 200-600m where there is also a high concentration of nitrate. The importance of nitrate at 200-600m depths is clearly more important. Tesdale has shown that the incomplete utilization of nitrate is really only important in the high latitudes and minimally influences the nitrate N15 at lower latitudes where that left over nitrate gets into the thermocline.

3. Statistical treatment of these records. How did the authors treat these multiple records with different proxies (foram bound or bulk?), locations, temporal resolutions? The authors state that FB-d15N is higher than bulk because FB-d15N neglects N2 fixation (I disagree, see my comments below), but when calculating the differences among these records, did the authors simply take the average or pick one proxy over the other?

The trends drawn through the data are 7-point moving averages (of the N-15 data as well as time) and we have now included the standard deviation on those lines. This was done using OriginPro and on combined datasets, so for example, all data for cores used to assemble the distal dataset were combined, same for deep, same for foram-bound. There are more differences between the sites reported by Kast et al., which is responsible for the greater standard deviation around 60 Ma. Even so, the “shapes” of the curves are the same. It is correct that the FB-N-15 is not always higher than the distal N-15, sometimes it is lower, for example during the Eocene. We have put more detail into the FB-record description (thank you for those comments).

Line 77. Clarify the processes that cause N to be lost preferentially over P in the marginal sea. Do the authors mean redfield uptake of N and P and burial in sediments, or denitrification in shallow sediments?

It was unclear, we referred to the loss of N through denitrification in the OMZ at boundary upwelling areas, not in sediments.

Line 87-89. The average d15N of ocean nitrate is determined by the ratio of N losses through water column or sedimentary denitrification. It is not determined by the upwelled nitrate along the margins. I do not know what the authors mean here. the marginal upwelling might affect the mean d15N value if upwelling drives sedimentary

denitrification, and assuming that water column denitrification remains constant through time. But the statement here made by the authors is wrong.

We admit this was poorly phrased and we have changed it.

Line 161-163. Nitrate d15N in the outer shelf should not equal deep ocean even if river influence has been removed in the mid and inner shelf.

We do not think this statement is correct. The on-welling of nitrate has been presented in the literature. The WOCE hydrographic sections show that along western boundaries (in the northern hemisphere, reverse for the southern) there is no severe O2 depletion and nitrate concentrations are quite uniform below 250m. We also present data at the shelf-slope margin at Cape Hatteras, a location where there is no OMZ, that shows bulk sediment $\delta^{15}\text{N}$ to be very similar to that of average deep nitrate. This is not the case for $\delta^{13}\text{C}$, which does show the effect of continental source of organic C.

Line 172-174. Foram-bound d15N may also be influenced by newly fixed N, when the fixed N goes into the food web. Overall FB-d15N shows strong correlation with thermocline nitrate, because insitu N2 fixation is a small flux comparing to vertical supplies. It is the same as sinking PON. The d15N of sinking PON, and thus buried bulk organic N in the sediments, should reflect the d15N of all the new sources of N to the euphotic zone. But comparing d15N of bulk sinking PN with these new sources, there is a stronger influence from vertical supplies, and not from insitu N2 fixation. So I would not say foram-bound d15N and bulk d15N differ in this.

This has been revised

Line 255, "where d15N of the ocean is defined". I guess the authors mean the deep ocean nitrate d15N. However, the deep ocean nitrate d15N is defined by the ratio between N loss occurring in the sediments vs. in the water column, not by the processes in the water column alone. This is a confusing sentence.

We refer the reviewer to our earlier comment

Figure 1. I think the locations of 1263 and 1209 are wrong.

They have been switched

Line 226. D15NFB captures the d15N signature of nitrate below the mixed layer, but

tends to neglect contributions from N fixation. See my previous comments. I do not think this is the case, at least there is no evidence for this statement.

This has been revised

The authors use $\delta^{15}\text{N}_{\text{distal}}$ and $\delta^{15}\text{N}_{\text{gyre}}$ to refer to the same thing, and use $\delta^{15}\text{N}_{\text{margin}}$ and $\delta^{15}\text{N}_{\text{deep}}$ to refer to the same thing, but need a statement at least when introducing the terms.

Some older terms were not captured in this revision, these have been fixed

Line 353. The lighter the oceanic N inventory becomes? I guess the authors mean the lighter the oceanic $\delta^{15}\text{N}$ becomes.

Correct, and this has been fixed.